# Prot2Text-V2: Protein Function Prediction with Multimodal Contrastive Alignment

**Xiao Fei[1], Michail Chatziannastasis[1], Sarah Almeida Carneiro[1], Hadi Abdine[2],**
**Lawrence P. Petalidis[3], Michalis Vazirgiannis[1,2]**
[1]École Polytechnique, Institut Polytechnique de Paris, France
[2]Mohamed bin Zayed University of Artificial Intelligence, United Arab Emirates
[3]M42 Health, United Arab Emirates
xiao.fei@polytechnique.edu
{almeidacarneiro, mvazirg}@lix.polytechnique.fr
mixalisx97@gmail.com, hadi.abdine@mbzuai.ac.ae, lpetalidis@m42.ae

## Abstract

Predicting protein function from sequence is a central challenge in computational biology. While existing methods rely heavily on structured ontologies or similarity-based techniques, they often lack the flexibility to express structure-free functional descriptions and novel biological functions. In this work, we introduce Prot2Text-V2, a novel multimodal sequence-to-text model that generates free-form natural language descriptions of protein function directly from amino acid sequences. Our method combines a protein language model as a sequence encoder (ESM-3B) and a decoder-only language model (LLaMA-3.1-8B-Instruct) through a lightweight nonlinear modality projector. A key innovation is our Hybrid Sequence-level Contrastive Alignment Learning (H-SCALE), which improves cross-modal learning by matching mean- and std-pooled protein embeddings with text representations via contrastive loss. After the alignment phase, we apply instruction-based fine-tuning using LoRA on the decoder to teach the model how to generate accurate protein function descriptions conditioned on the protein sequence. We train Prot2Text-V2 on about 250K curated entries from SwissProt and evaluate it under low-homology conditions, where test sequences have low similarity with training samples. Prot2Text-V2 consistently outperforms traditional and LLM-based baselines across various metrics.

## 1 Introduction

As sequencing technologies uncover millions of new proteins, most remain uncharacterized, underscoring the urgent need for accurate, scalable function prediction. Traditional tools like BLAST [Altschul et al., 1990] and InterPro [Hunter et al., 2009], though effective, rely on manual curation and homology-based inference, making them slow and limited—especially for proteins from poorly studied organisms lacking close homologues. These approaches are also constrained by predefined labels or controlled vocabularies, restricting their ability to express the full complexity of protein function. Protein function is complex, spanning molecular mechanisms, pathways, and evolutionary roles. Free-text predictions, enabled by natural language generation, offer greater flexibility and expressiveness than rigid labels like Gene Ontology (GO) terms. This has spurred growing interest in models that can generate rich, compositional natural language annotations of protein function.

Although deep learning models for protein function prediction have achieved promising performance, usually they have not been thoroughly tested on protein samples with low sequence similarity

---

[1]The source code for this project is available at https://github.com/ColinFX/Prot2Text-V2/

39th Conference on Neural Information Processing Systems (NeurIPS 2025).

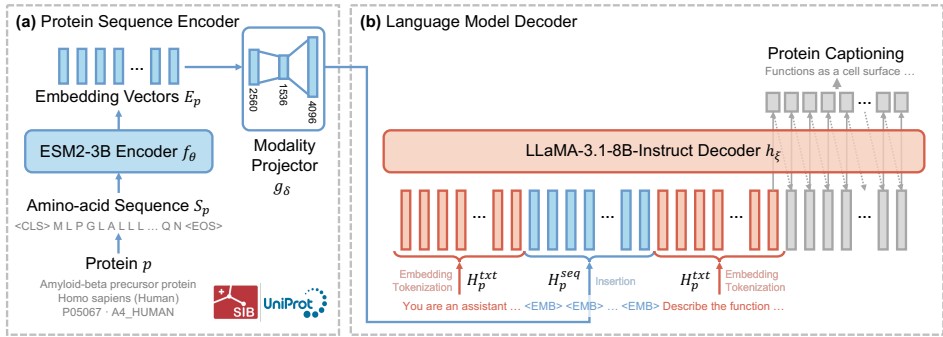

Figure 1: Illustration of the multimodal architecture and primary data flow of Prot2Text-V2. **(a)** In the protein encoder, the amino-acid sequence is first encoded by the protein language model and then projected to the hidden dimension of the text decoder with a two-layer non-linear modality projector. **(b)** In the language model decoder, the projected sequence of protein embeddings is inserted into the sequence of word embeddings of the user message, forming the prompt to the language model. Finally, the decoder generates the description of the protein auto-regressively.

to training samples, which is a critical scenario for real-world generalization. This gap raises concerns that existing models may not truly understand protein function, but instead rely heavily on sequence similarity. As a result, they often fail to generalize beyond their training data, limiting their effectiveness for novel or poorly characterized proteins. To address this, we propose Prot2Text-V2, a multimodal sequence-to-text framework that generates rich, textual functional descriptions directly from amino acid sequences. Our approach combines a powerful protein encoder (ESM-3B [Lin et al., 2023]) with a large language model decoder (LLaMA-3.1-8B-Instruct [Grattafiori et al., 2024]).

A key innovation of our approach is the Hybrid Sequence-level Contrastive Alignment Learning (H-SCALE) framework, which establishes a direct and parameter-efficient alignment between protein and text representations. Unlike prior models such as ProtT3 that rely on additional attention modules (e.g., Q-Formers) and redundant text encoders to achieve multimodal alignment, H-SCALE performs explicit contrastive alignment between projected protein embeddings and intermediate text embeddings within the decoder itself. This design preserves the full expressivity of the pretrained protein encoder, removes the learned bottleneck introduced by query-based alignment, and significantly reduces latency and memory overhead. Following this alignment phase, the decoder is fine-tuned using instruction-based LoRA [Hu et al., 2022], enabling controlled natural language generation conditioned on protein features and prompts. To comprehensively assess the model, we evaluate Prot2Text-V2 on low-homology proteins, where accurate generalization beyond sequence similarity is most biologically meaningful. Our evaluation includes automated metrics, LLM-as-a-judge scoring for semantic fidelity, and expert human assessment, providing a multi-faceted understanding of model quality and robustness.

Our contributions can be summarized as follows:

- We introduce **Prot2Text-V2**, a model that generates human-readable descriptions of protein function directly from amino acid sequences. It combines a pretrained ESM-3B protein encoder with an instruction-tuned LLaMA-3.1-8B decoder. We combine them via a **two-layer nonlinear modality projector**, which simplifies training, accelerates inference, and avoids the need for complex cross-attention mechanisms used in prior work.
- We introduce **H-SCALE**, a **new contrastive learning scheme** based on both mean and standard deviation–pooled sequence embeddings and intermediate decoder text representations. This alignment phase enables more effective downstream instruction tuning.
- We highlight Prot2Text-V2's **strong performance in the weak-homology regime**, where generalization is most critical and biologically relevant. We achieve state-of-the-art results across lexical and semantic metrics, outperforming both traditional methods and recent LLM baselines.

The rest of the paper is structured as follows. Section 2 reviews related work on protein function prediction, multimodal learning, and contrastive learning. Section 3 details our proposed method, Prot2Text-V2, including its architecture, the contrastive alignment strategy, and instruction-based fine-

tuning. Section 4 covers the experimental setup, datasets, baselines, and results. Section 5 discusses limitations and future directions, and Section 6 concludes. Additional materials are provided in the Appendix.

## 2 Related Work

**Traditional Protein Function Prediction:** Protein function annotation has traditionally relied on sequence homology and structured ontologies like Gene Ontology (GO). Early tools such as BLAST [Altschul et al., 1990] and Foldseek [van Kempen et al., 2022] identify similar proteins to infer function, but they struggle with novel sequences lacking close homologs. To address this, recent deep learning approaches predict GO terms from sequence and/or structure using convolutional networks, graph-based models, and protein language models [Kulmanov and Hoehndorf, 2020, Yuan et al., 2024, 2023, Fan et al., 2020, Li et al., 2023]. More recent efforts apply autoregressive modeling, prompt tuning, and hybrid architectures to refine label prediction [Wang et al., 2025, Rong et al., 2024]. While these methods improve scalability and coverage, they still reduce complex protein functions to predefined labels, limiting the granularity and expressiveness of the annotations.

**Protein-to-Text Generative Models:** Recent approaches have shifted toward generating natural language descriptions directly from protein sequences, offering a more flexible alternative to predefined labels. Models like Prot2Text [Abdine et al., 2024] pioneered this direction but still face challenges in effectively aligning sequence encoders with language decoders. Subsequent work has explored a range of strategies: some focus on sequence- or structure-based generation [Liu et al., 2024b, Xiao et al., 2024], others leverage multimodal or iterative training across molecular representations [Liu et al., 2024a, Zhou et al., 2025, Kim et al., 2025, Wang et al., 2023], and recent models incorporate instruction tuning and large language models for functional synthesis [Lv et al., 2025, Wang et al., 2024a].

**Contrastive Protein-Text Alignment:** Contrastive learning has emerged as a powerful strategy for aligning protein sequences with textual annotations, enabling richer multimodal representations. Early work adapted CLIP-style frameworks to protein embeddings with separate text encoders [Wu et al., 2024, Radford et al., 2021], while subsequent models incorporated structural and functional modalities to enhance alignment [Su et al., 2024, Liu et al., 2024b, Ünsal et al., 2024, Wang et al., 2024b]. Transformer-based methods with contrastive objectives [Zheng and Li, 2024, Xu et al., 2023] further improve alignment by leveraging sequence-aware architectures. Despite these advances, many models still depend on pooled or special-token embeddings, which can miss fine-grained details.

## 3 Methodology

The major objective of Prot2Text-V2 is to generate accurate and informative free-text descriptions of protein function directly from amino acid sequences. This task requires a model capable of understanding and aligning protein representations with natural language. However, LLMs operate in a different semantic space that is not natively aligned with protein representations. To address this challenge, our method introduces: (1) a lightweight and flexible decoder-only architecture with a nonlinear modality projector and (2) a brief yet effective contrastive alignment phase before standard instruction tuning.

### 3.1 Decoder-Only Model Architecture

Protein language models (PLMs) like Evolutionary Scale Modeling-2 (ESM2) [Lin et al., 2023] have demonstrated a strong ability to capture evolutionary signals, structural features, and functional characteristics by leveraging large-scale pretraining on millions of protein sequences. Therefore, we choose ESM2 as the protein amino-acid sequence encoder $f_\theta(\cdot)$ with pre-trained parameters $\theta$. For an input protein amino-acid sequence $S_p \in \mathbb{N}^l$ composed of a series of amino-acid tokens, we first embed it with the ESM2-3B encoder, which returns a sequence of updated embedding vectors as sequence features: $E_p = f_\theta(S_p) \in \mathbb{R}^{l \times d_f}$. We then employ LLaMA-3.1-8B-Instruct [Grattafiori et al., 2024] as our foundation large language model $g_\delta$ to tackle with the protein captioning task. This LLM has been extensively pretrained on diverse textual corpora and further tuned for instruction-following tasks, making it well-suited for our task.

The protein encoder and language decoder differ in both hidden dimensions and semantic spaces, making them incompatible. Directly injecting raw protein embeddings into the decoder can result in semantic misalignment, degrading generation quality. Prior models [Brandes et al., 2022, Rao et al., 2019] address this with cross-attention or multimodal transformers, but require extensive data and training. To bridge this gap, we introduce a two-layer nonlinear projection module $h_\xi$, as in Equation 1, that maps protein embeddings into the decoder's semantic space. It uses GELU activation [Hendrycks and Gimpel, 2016] for nonlinearity and layer normalization (LN) to stabilize training, producing decoder-compatible embeddings for effective cross-modal conditioning.

$$H_p^{\text{seq}} = h_\xi(E_p) = \text{LN}(W_2 \cdot \sigma(W_1 \cdot E_p)) \in \mathbb{R}^{l \times d_g}, \text{ with } W_1 \in \mathbb{R}^{d_g \times d_f}, W_2 \in \mathbb{R}^{d_g \times d_g} \quad (1)$$

Finally, we incorporate the projected embedding sequence into the language decoder's chat dialogue using the following structured template with a flexible user message. The assistant's response (colored in red) is left for the model to auto-regressively generate:

---

**System**: You are a scientific assistant specialized in protein function predictions. Given the sequence embeddings of a protein, describe its function clearly and concisely in professional language.
**User**: (Protein name: `<NAME>`;)* (taxonomy: `<TAXON>`;)* sequence embeddings: $H_{p,1}^{\text{seq}}|H_{p,2}^{\text{seq}}|...|H_{p,l}^{\text{seq}}$
**Assistant**: `<PROTEIN DESCRIPTION>`
* Optional metadata text fields

---

Figure 2: A flexible chat template for language decoders, where user message metadata fields are subject to dropout during training. During evaluation, these fields are either fully retained or completely removed, enabling two distinct testing scenarios.

Our integration approach specifically inserts placeholder tokens into the chat dialogue, which the language decoder initially converts to word embeddings $H_p^{\text{txt}}$. We then perform a soft-prompt [Lester et al., 2021] replacement of the protein sequence segment with the encoder-adapter's output embeddings. This direct integration strategy - unlike methods like [van den Oord et al., 2017] requiring additional discretization steps - provides two key advantages: (1) it preserves the full semantic richness of the encoder's continuous representations, and (2) enables precise multimodal alignment by eliminating information bottlenecks that typically occur in codebook-based approaches.

### 3.2 Two-Stage Training Process

We propose a two-stage alignment and fine-tuning approach to first align and project the protein sequence encoder's outputs $E_p$ to the same hidden semantic space as the language decoder, then fine-tune the decoder to produce accurate functional descriptions by conditioning on the aligned protein representations. The additional alignment stage ensures stable cross-modal understanding before the standard end-to-end fine-tuning, thus producing reliable predictions by leveraging the full context of encoder hidden states.

#### 3.2.1 Hybrid Sequence-Level Contrastive Alignment Learning

In this alignment stage, we connect our proposed modality adapter to the protein sequence encoder and train the system using sequence-description pairs $(S_p, T_p)$ from our training set (illustrated in Figure 3). The amino-acid sequence will be processed following a standard pipeline, first by the pre-trained sequence encoder and then the randomly initialized modality adapter (Equation 2). In parallel, we process the protein's text description through the frozen language decoder, which comprises 32 self-attention blocks, and extract the intermediate representations from the 16th block (Equation 3). Where $g = g_{\text{head}} \circ g_{32} \circ g_{31} \circ \ldots \circ g_1 \circ g_{\text{embed}}$ denote the composition of the sequential self-attention blocks in the LLaMA-3.1 architecture, including the language model head and the input token embedding layer. The sequence-level protein embedding is represented as $H_{p,i}^{\text{seq}} \in \mathbb{R}^{d_h}$, where $d_h = d_g$ by projection. Moreover, for most protein–text pairs, the sequence length $l_{\text{seq}}$ is not equal to the text length $l_{\text{txt}}$, i.e., $l_{\text{seq}} \neq l_{\text{txt}}$.

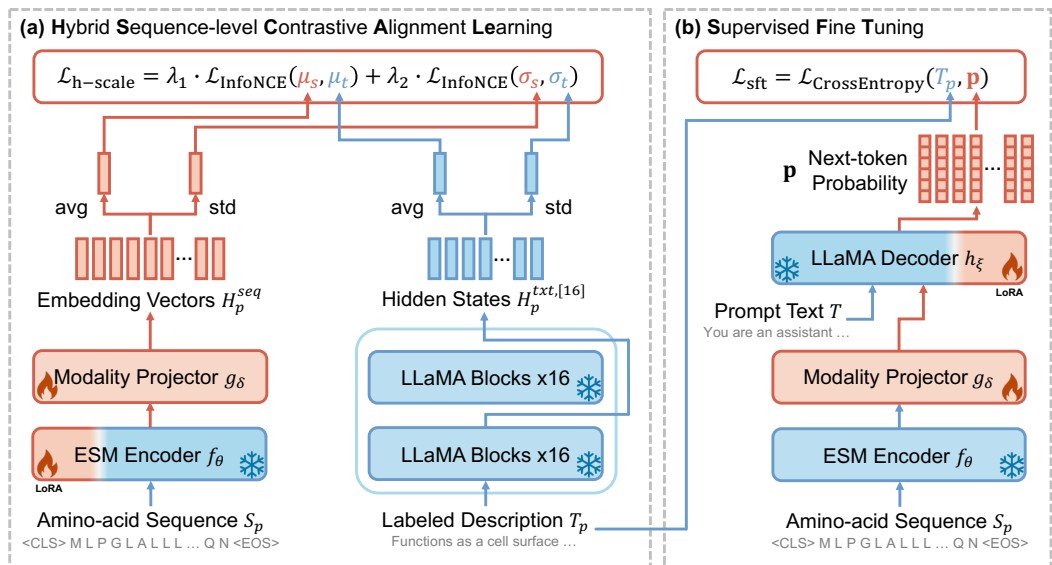

Figure 3: Illustration of the two-stage training process of Prot2Text-V2. **(a)** During the first stage, Hybrid Sequence-level Contrastive Alignment Learning (H-SCALE) aligns the protein encoder outputs with the semantic space of the pretrained language decoder, enabling effective cross-modal learning. **(b)** In the second stage, instruction-based supervised fine-tuning is performed to teach the decoder how to generate protein function descriptions based on the aligned embeddings. Red blocks denote trainable parameters, while blue blocks indicate frozen components.

$$H_p^{\text{seq}} = h_\xi(f_\theta(S_p)) = [H_{p,1}^{seq}, \ldots, H_{p,l_{seq}}^{seq}], \text{ where } H_{p,i}^{seq} \in \mathbb{R}^{d_g} \tag{2}$$

$$H_p^{\text{txt},[16]} = g_{16}(g_{15}(...(g_1(g_{\text{embed}}(T_p)))...)) = [H_{p,1}^{txt}, \ldots, H_{p,l_{txt}}^{txt}] \tag{3}$$

This strategy overcomes two key issues in aligning a language decoder with a different-modality encoder: (1) raw input token embeddings lack the self-attention context needed to capture linguistic relationships, and (2) final decoder outputs are overspecialized for next-token prediction, losing broader semantic information through extensive autoregressive optimization. By using the 16th decoder block, we capture rich contextualized representations that better encode protein-relevant semantics. The extracted representations are aligned with the protein sequence embeddings via a combined InfoNCE contrastive loss [Oord et al., 2018], applied to both mean- and standard-deviation-pooled features:

$$\mathcal{L}_{\text{align}} = \lambda_1 \, \mathcal{L}_{\text{InfoNCE}}(\mu_s, \, \mu_t) + \lambda_2 \, \mathcal{L}_{\text{InfoNCE}}(\sigma_s, \, \sigma_t), \tag{4}$$

$$\text{where} \quad \mathcal{L}_{\text{InfoNCE}}(x, \, y) = -\log \frac{\exp(x^\top y/\tau)}{\sum_{i=1}^{B} \exp(x^\top y_i/\tau)}, \tag{5}$$

where $\mu_s = \text{MeanPool}(H_p^{\text{seq}})$ and $\mu_t = \text{MeanPool}(H_p^{\text{txt},[16]})$ denote the global mean-pooled embeddings of the protein sequence and the labeled description, respectively. Similarly, the textual features are pooled to obtain $\sigma_s = \text{MeanPool}(H_p^{\text{seq}})$ and $\sigma_t = \text{StdPool}(H_p^{\text{txt},[16]})$, capturing local variation through mean and standard deviation pooling. The temperature hyperparameter is set to $\tau = 0.07$, and the batch size is $B = 1024$. Finally, the loss terms are balanced using weights $\lambda_1 = 0.7$ and $\lambda_2 = 0.3$, which control the contributions of global versus local alignment.

By minimizing $\mathcal{L}_{\text{align}}$, we enforce both global consistency and local variation alignment between the original and transformed embeddings, resulting in more robust cross-modal representations. This alignment phase, performed before standard end-to-end fine-tuning, plays a critical role: it projects the protein embeddings into the same semantic space as the decoder's token representations. As a result, the decoder can more effectively integrate information from both the input prompt and

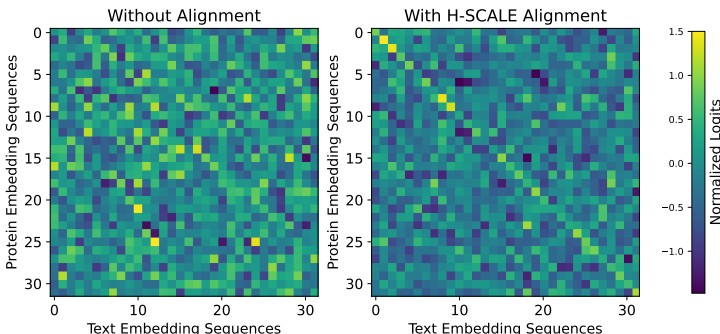

Figure 4: Illustration on similarity matrices before (left) and after (right) alignment on a group of test protein-description pairs. After alignment, protein embeddings show strong one-to-one correspondence with their matching text embeddings, as indicated by the prominent diagonal. This reflects significantly higher similarity between each protein sequence embedding and its own text description compared to mismatched pairs, demonstrating successful cross-modal alignment.

the protein features. This reduces the risk of representational mismatch, where the decoder might otherwise ignore or overemphasize one modality due to incompatible embedding distributions.

We visualize the effect of the contrastive learning on pairwise similarity between protein embeddings and their corresponding text embeddings in Figure 4. Before applying H-SCALE, the similarity matrix is diffuse and lacks a clear structure, suggesting that protein and text embeddings are poorly aligned, as they show high similarity with mismatched pairs. After our contrastive alignment, the matrix exhibits strong diagonal dominance, where each protein sequence embedding is most similar to its true text description. Further details are also explored in Appendix A.

### 3.2.2 End-to-End Fine-Tuning

In contrast to the alignment stage where we fix the language decoder then train the modality adapter and partially the protein encoder, in this stage, we focus on fine-tuning the pretrained language decoder with Low Rank Approximation (LoRA) [Hu et al., 2022] with an end-to-end teacher forcing cross entropy to regulate the conditional auto-regressive generation of the decoder.

The model is fine-tuned on instruction-formatted protein-caption pairs, where each input consists of (1) a protein sequence encoded as multimodal embeddings $H_p^{seq}$, (2) optional metadata including the name of the protein and its taxonomic classification, and (3) a task instruction $T$ guiding the model to generate the description. The training objective maximizes the likelihood of the target caption while masking loss computation over the input prompt tokens. Specifically, given an input sequence embeddings $H$ and target tokens $T_p$, the model minimizes the supervised fine tuning loss:

$$\mathcal{L}_{sft} = \mathcal{L}_{\text{CrossEntropy}}(T_p, p) = -\sum_{t=1}^{T} \log p\big(t_p | H, T_{p,<t}\big) \tag{6}$$

where $p$ represents the predicted next-token probability distribution using the language model head of the decoder. The gradients are propagated only over the caption tokens $T_p$, ensuring the model learns to generate accurate descriptions without overfitting to the prompt structure.

To standardize the input, the structured chat template is applied to integrate system instructions, user messages and the response of the model. This ensures consistent formatting across diverse prompts, improving the generalization ability of the model. To improve robustness, we apply dropout to each metadata field (name and taxonomy) with an 80% probability during training. This approach addresses incomplete inputs, reduces over-reliance on potentially noisy annotations, and enhances generalization by simulating real-world scenarios where metadata may be missing or inconsistent. This supervised fine-tuning stage further aligns the decoder's generative behavior with protein

Table 1: Evaluation results for Prot2Text-V2 and baselines. Lexical metrics include exact match ratio (Exct), BLEU-2 F1 Score (B-2), BLEU-4 F1 Score (B-4), ROUGE-1 F1 Score (R-1) , ROUGE-2 F1 Score (R-2), and ROUGE-L F1 Score (R-L). Semantic metrics comprise BERTScore using RoBERTa (RBT) and BioBERT (BBT) embeddings, plus a GPT-4o-based LLM judge score (GPT). Prot2Text-V2 with H-SCALE outperforms general-purpose LLMs, as well as various baselines and contrastive learning variants.

| Model | Exct | B-2 | B-4 | R-1 | R-2 | R-L | RBT | BBT | GPT |
|---|---|---|---|---|---|---|---|---|---|
| GPT-4o-mini [OpenAI, 2023](0-shot) | - | 2.65 | 0.31 | 10.81 | 1.25 | 7.73 | 81.44 | 71.36 | - |
| GPT-4o-mini(1-shot) | - | 5.86 | 1.24 | 15.38 | 2.66 | 11.58 | 83.21 | 72.39 | - |
| Claude-3.5-Sonnet [Anthropic, 2024] (0-shot) | - | 5.47 | 0.90 | 13.91 | 1.92 | 10.22 | 82.98 | 72.39 | - |
| Claude-3.5-Sonnet(1-shot) | - | 6.62 | 1.54 | 16.02 | 3.09 | 12.14 | 83.58 | 71.79 | - |
| P2T-GPT [Zhang et al., 2024] | 17.24 | 33.30 | 26.06 | 38.14 | 23.72 | 35.47 | 88.23 | 78.66 | - |
| Prot2Text [Abdine et al., 2024] | 32.05 | 36.36 | 32.45 | 50.49 | 42.60 | 48.33 | 90.56 | 84.26 | 56.95 |
| Transformer [Vaswani et al., 2017] | - | 17.48 | 15.75 | 27.80 | 19.44 | 26.07 | 82.31 | 75.58 | - |
| ESM2GPT | - | 35.49 | 32.11 | 47.46 | 39.18 | 45.31 | 89.97 | 83.21 | - |
| RGCN2Transformer | - | 31.64 | 27.97 | 42.43 | 34.91 | 40.72 | 90.58 | 84.30 | - |
| RGCN2GPT | - | 24.05 | 21.63 | 36.20 | 28.01 | 34.40 | 85.12 | 78.91 | - |
| RGCNxESM2GPT | - | 33.01 | 30.39 | 45.75 | 37.38 | 43.63 | 89.04 | 82.51 | - |
| SEQ2LLaMA | 20.85 | 24.97 | 21.44 | 33.99 | 25.86 | 32.18 | 87.13 | 77.32 | 30.99 |
| RGCN2LLaMA | 22.79 | 25.96 | 22.46 | 39.36 | 31.34 | 37.60 | 88.31 | 79.58 | 30.69 |
| RGCNxESM2LLaMA | 30.07 | 35.84 | 31.84 | 50.87 | 43.28 | 48.98 | 90.92 | 84.75 | 58.48 |
| Prot2Text-V2 (w/o H-SCALE) | 36.52 | 42.70 | 38.98 | 54.71 | 47.62 | 52.90 | 91.33 | 85.58 | 60.93 |
| **Prot2Text-V2 (w/ H-SCALE)** | 39.16 | 46.67 | 43.34 | 57.24 | 50.17 | 55.39 | 91.95 | 86.81 | 64.58 |

captioning through structured instruction tuning. Finally, we discuss the computational complexity of the whole pipeline in Appendix B.

# 4 Experiments and Results

## 4.1 Experimental Setup

**Dataset:** The dataset used during our experiments was the one proposed by Abdine et al. [2024]. This dataset is a multimodal dataset with 256,690 proteins, filtered originally from SwissProt (see Appendix C) [Bairoch and Apweiler, 2000], with their respective corresponding sequence, the predicted structure from AlphaFold [Jumper et al., 2021], and the textual description. This curated dataset provides aligned sequence, structure, and text modalities, with rigorous filtering and low redundancy across splits, making it better suited for our multimodal learning tasks. Notably, to the best of our knowledge, this is the only publicly available dataset that minimizes protein sequence similarity between training and test sets. We further discuss our data splits in Appendix C.1.

**Hyperparameters:** Our model is implemented using PyTorch and trained on a single node with 8 NVIDIA A100 80GB GPUs. We use the AdamW optimizer [Loshchilov and Hutter, 2017] with $\epsilon = 1 \times 10^{-6}$, $\beta_1 = 0.9$, and $\beta_2 = 0.999$. The learning rate starts at $0.0002$ and decays to zero following a cosine scheduler, with a warm-up period covering $6\%$ of the total training steps. For the LoRA adapter, we apply it to the self-attention modules in both the ESM encoder and LLaMA decoder, using a rank of $32$ and an $\alpha$ value of $64$. Training lasts for 12 epochs in contrastive learning and 24 epochs for supervised fine-tuning. During the contrastive learning stage, the batch size per device is $1024$, which is further divided into 8 chunks to accommodate memory constraints. The batch size is set to 4 per GPU, and gradient accumulation is applied every 8 forward passes, resulting in an effective batch size of 256.

**Evaluation Metrics:** For quantitative evaluation, we assess Prot2Text-V2 using standard sequence generation metrics: BLEU [Papineni et al., 2002], ROUGE (1, 2, L) [Lin, 2004], BERTScore [Zhang et al., 2019], and contextual metrics based on RoBERTa [Liu et al., 2019] and BioBERT [Lee et al., 2020]. BLEU and ROUGE capture lexical overlap—BLEU emphasizing precision and ROUGE recall—while BERT-based metrics evaluate semantic similarity using contextual embeddings. As

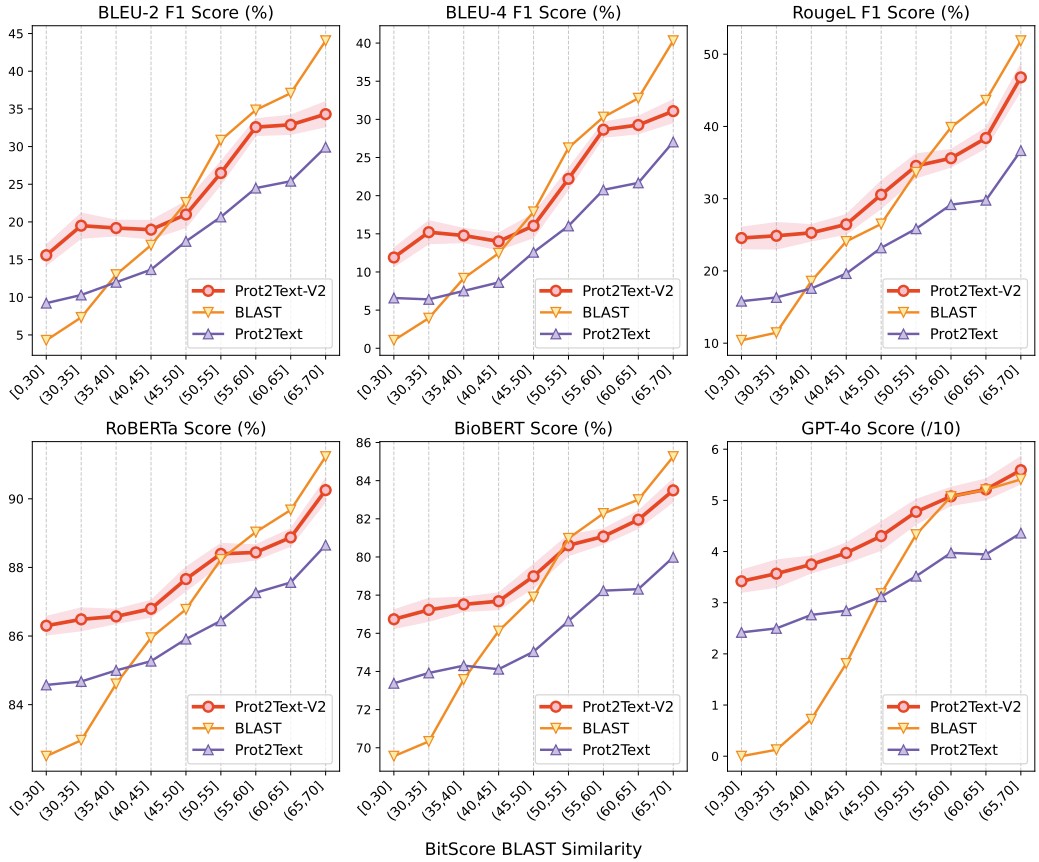

Figure 5: Evaluation across BLAST sequence bitscore bins (%) shows that Prot2Text-V2 outperform baselines (Prot2Text and BLAST). The evaluations show consistent improvements, particularly in the [0–50] bins, highlighting better generalization to remote homologs.

BLEU and ROUGE are non-decomposable corpus-level scores, we estimate their variance and confidence intervals using the method described in Appendix E. Additionally, we include a semantic evaluation metric using GPT-4-mini as an LLM-based judge. Unlike traditional metrics, it assesses prediction correctness by reasoning about semantic content, accommodating functional variations and omissions due to token truncation. This allows more flexible evaluation, particularly when predictions partially align with or generalize beyond the ground truth. Full details are in Appendix H. **Baseline models and ablations** We evaluate our model against methods from two distinct paradigms: traditional sequence-alignment-based methods and deep learning generative methods. In the following, we describe the baselines in more detail.

**BLAST:** BLAST [Altschul et al., 1990] is a classic alignment-based tool that predicts protein function by finding homologs with known annotations. It works well for high sequence similarity (>70%) but struggles below 40%, limiting its effectiveness for novel or distant proteins. *Despite this, BLAST is often omitted from recent LLM-based function prediction comparisons, making it difficult to assess whether neural models genuinely generalize or mainly retrieve similar sequences.*

**Prot2Text:** We also compare Prot2Text-V2 against Prot2Text and a range of unimodal and multimodal baselines based on the cross-attention framework of Abdine et al. [2024]. Unimodal models—RGCN [Schlichtkrull et al., 2018], ESM, and a Transformer [Vaswani et al., 2017] trained from scratch—process either structural graphs or sequences independently. Multimodal baselines include RGCN+ESM, which concatenates graph and sequence embeddings without explicit amino-acid level alignment, and RGCN combined with a scratch-trained Transformer. We further evaluate multiple sizes of Prot2Text, all using GPT-2 [Radford et al., 2018] as the language decoder.

**ProtT3:** We compare our method to ProtT3 [Liu et al., 2024b], a state-of-the-art approach that also employs a language model with ESM-2 and a cross-modal projector for protein encoding. While

there are some similarities with ProtT3 architecture, our model departs in the following essential respects: ProtT3 relies on a Q-Former, an additional attention module with learnable queries of a fixed number and a redundant text encoder, to align the protein and text modalities. Which introduces computational overhead, potential information loss from query transformation, and lacks a thorough evaluation of weak homology proteins. In contrast, Prot2Text-V2 follows a decoder-only design with a lightweight projector and with a parameter-free alignment method. Rather than learning alignment implicitly through a query-attention module, H-SCALE directly aligns the projected protein representations to intermediate text embeddings with a new contrastive learning objective. This eliminates the learned bottleneck, preserves the full expressivity of the pretrained protein encoder, and avoids added parameters, latency, and memory, and also allows us to use a more powerful LLM decoder.

**P2T-GPT:** We also compare our model to P2T-GPT [Zhang et al., 2024] , a relatively lightweight model (160M parameters) trained entirely from scratch without leveraging large-scale pretrained components or proper multimodal alignment.

**Generic LLMs:** Despite not being specifically designed for protein analysis, general-purpose LLMs such as Galactica [Taylor et al., 2022], Claude 3.5 Sonnet [Anthropic, 2024], LLaMA 3.1 [Grattafiori et al., 2024], and GPT-4o mini [OpenAI, 2023] have been trained on massive and diverse datasets and are widely adopted in practice. We include these models in our comparison to assess whether our specialized pretraining approach offers superior performance compared to zero-shot and few-shot predictions from LLMs.

**Ablation Studies:** To validate the effectiveness of our architecture, we conduct ablation studies across five variants: (i) SEQ2LLaMA, where raw amino acid sequences (in text format) are inserted directly into the user prompt for the model; (ii) RGCN2LLaMA, using an RGCN encoder to encode protein structure predicted with AlphaFold [Varadi et al., 2022]; (iii) RGCNxESM2LLaMA, which fuses outputs from both RGCN and ESM encoders via a gated cross-attention fusion module to integrate structural and sequence information; and (iv) Prot2Text-V2 (w/o CL), which skips contrastive pretraining and begins directly from instruction tuning. These variations allow us to isolate the contributions of individual components.

## 4.2 Results

We present our evaluation results in protein function prediction in Table 1. Prot2Text-V2 (w/ H-SCALE) outperforms all baselines across lexical and semantic metrics. It achieves top BLEU-2 (48.32) and BLEU-4 (36.07) scores, reflecting strong n-gram precision, along with leading ROUGE-1 (63.51), ROUGE-2 (45.06), and ROUGE-L (59.66) scores, indicating improved fluency and recall. For semantic alignment, it surpasses general-purpose LLMs (RBT: 83.58), Prot2Text variants (RBT: 90.58), and ablations (RBT: 91.33), reaching a top RoBERTa score of 91.95. On the domain-specific BioBERT metric, it leads with 86.81, outperforming general LLMs (72.39), prior variants (84.30), and ablations (85.58). Prot2Text-V2 (w/ H-SCALE) also excels in semantic alignment with reference texts. Its GPT score of 64.58 further confirms superior semantic coherence and generation quality, demonstrating strong lexical accuracy and contextual relevance for robust text generation.

Furthermore, we evaluate performance focusing on the challenging low-similarity cases where traditional methods often fail. Figure 5 shows average BLEU-2, BLEU-4, ROUGE, RoBERTa, BioBERT, and GPT-4o judge metrics across BLAST bitscore bins ranging from low (0–30) to medium-high (65–70). While BLAST outperforms multimodal methods at higher similarity levels (above 55%), it performs poorly in low-similarity ranges (0–50%) where sequence copying is ineffective. Notably, Prot2Text-V2 achieves the highest scores in these low-similarity scenarios. Additional details on BLAST score computation are provided in Appendix D.

We further compare our main model, Prot2Text-V2 (w/ H-SCALE), against ProtT3 [Liu et al., 2024b] and its variants in Table 2. We retrained Prot2Text-V2 using the same dataset split provided by ProtT3, which follows a random partitioning strategy and does not explicitly focus on the low sequence similarity regime. Additionally, to evaluate the models under the same low-homology settings, in Tables 3, and 4, we filtered test proteins with the same bitscore ranges as shown in Figure 5, and made a direct comparison between different approaches on the same ranges to evaluate BLEU-4 F1, and BioBert F1 scores. Prot2Text-V2 consistently outperforms ProtT3 across the reported metrics, highlighting the superior performance of our method under their evaluation protocol. From the comparison with P2T-GPT, we also conclude that Prot2Text-V2 benefits from integrating powerful pretrained models (ESM and LLaMA) and an H-SCALE adapter, enabling it to capture richer biological and linguistic priors. This architectural difference is reflected in the substantial performance

gains across all evaluation metrics, especially in semantic fidelity and functional relevance. Finally, we report extra qualitative evaluations along with human expert evaluations in Appendix F and G.

Table 2: Evaluation results of our proposed model against ProtT3 and Galactica. For a fair comparison, we report results on the original data split from ProtT3 paper.

| Model | Exct | B-2 | B-4 | R-1 | R-2 | R-L | M-R |
|---|---|---|---|---|---|---|---|
| Galactica [Taylor et al., 2022] | 13.49 | 42.48 | 37.79 | 44.42 | 34.73 | 42.46 | 48.27 |
| ProtT3 (w/ MLP) | 20.60 | 48.96 | 44.59 | 57.28 | 50.17 | 56.89 | 57.30 |
| ProtT3 (w/o stage 1) | 22.88 | 50.21 | 46.76 | 58.64 | 51.63 | 57.17 | 58.62 |
| ProtT3 [Liu et al., 2024b] | 25.74 | 55.03 | 51.47 | 63.67 | 56.59 | 62.16 | 63.63 |
| **Prot2Text-V2** | **49.04** | **60.16** | **57.29** | **67.33** | **61.39** | **65.61** | **66.05** |

Table 3: BLEU-4 (B-4) F1 scores of Prot2Text, ProtT3, and Prot2Text-V2 across BLAST sequence bitscore bins (%).

| Model/BitScore | [0,30] | (30,35] | (35,40] | (40,45] | (45,50] | (50,55] | (55,60] | (60,65] | (65,70] |
|---|---|---|---|---|---|---|---|---|---|
| Prot2Text | 6.58 | 6.41 | 7.51 | 8.63 | 12.58 | 16.03 | 20.76 | 21.68 | 27.04 |
| ProtT3 | 6.19 | 5.26 | 7.12 | 9.76 | 13.10 | 10.87 | 18.97 | **30.95** | **40.82** |
| Prot2Text-V2 | **11.89** | **15.21** | **14.78** | **14.00** | **16.05** | **22.20** | **28.65** | 29.26 | 31.08 |

Table 4: BioBERT (BBT) F1 scores of Prot2Text, ProtT3, and Prot2Text-V2 across BLAST sequence bitscore bins (%).

| Model/BitScore | [0,30] | (30,35] | (35,40] | (40,45] | (45,50] | (50,55] | (55,60] | (60,65] | (65,70] |
|---|---|---|---|---|---|---|---|---|---|
| Prot2Text | 73.38 | 73.92 | 74.30 | 74.11 | 75.04 | 76.64 | 78.24 | 78.31 | 78.98 |
| ProtT3 | 67.57 | 66.44 | 69.19 | 74.37 | **79.73** | 78.11 | 76.44 | 81.06 | **83.99** |
| Prot2Text-V2 | **76.74** | **77.23** | **77.52** | **77.68** | 78.99 | **80.61** | **81.07** | **81.95** | 83.49 |

## 5 Limitations and Future Work

While our evaluation shows promising results using lexical and semantic metrics, these may not fully reflect functional correctness. Therefore, a limitation of the work is the lack of experimental (wet-lab) validation to confirm biological accuracy. For future work, we plan to establish collaborations with experimental biologists to carry out wet-lab validation of selected proteins, enabling more biologically grounded evaluation. Furthermore, future work will focus on integrating models with stronger reasoning capabilities that can better capture the protein function. We also aim to improve our model by incorporating additional molecular modalities, such as structural and surface features.

## 6 Conclusion

In this work, we proposed Prot2Text-V2, a novel multimodal framework for the prediction of protein function in free text format. First, we introduce a contrastive learning pretraining strategy that aligns the protein sequence representation with the text embeddings. Then, we leveraged instruction-based fine-tuning to teach the decoder how to condition on the aligned protein embeddings and generate accurate functional descriptions in natural language. Through extensive evaluation under challenging weak-homology conditions, Prot2Text-V2 consistently outperformed traditional alignment methods, prior multimodal approaches, and general-purpose LLMs on both lexical and semantic metrics. Our ablation studies further confirmed the importance of the contrastive alignment phase and the design choices in our architecture. We believe that Prot2Text-V2 represents a significant step toward scalable and flexible protein annotation, with applications in functional genomics, drug discovery, and synthetic biology.

# 7 Acknowledgments

We would like to thank the anonymous reviewers of the 39th NeurIPS conference for their valuable comments and constructive suggestions. We are also grateful to Dr. Johannes Lutzeyer for his assistance with the mathematical aspects during the rebuttal process. Our sincere thanks go to Dr. Lawrence P. Petalidis, Dr. Maria Dimitriadi, and Aron Vukatana for providing professional guidance on the subject and expert evaluation of the results. This work was granted access to the HPC resources of IDRIS made available by GENCI, and the computations were further supported by the Berzelius resource provided by the Knut and Alice Wallenberg Foundation at the National Supercomputer Centre.

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

# A   Experimental Details on H-SCALE

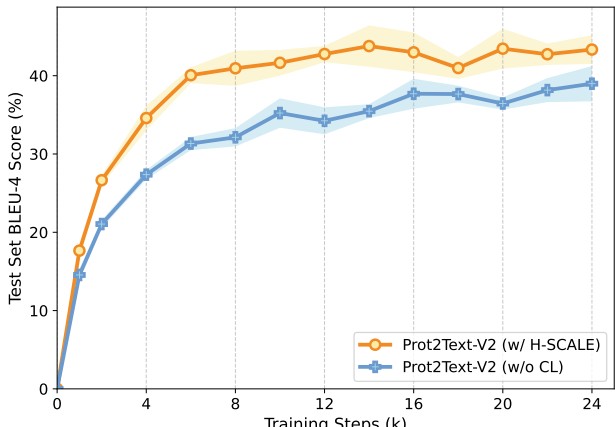

Figure 6: BLEU-4 scores on the test set plotted over training steps plot for two Prot2Text-V2 variants: with H-SCALE and without contrastive learning (w/o CL). These results are from supervised fine-tuning over approximately 48 hours. However, our H-SCALE contrastive learning converges in just 10 hours on the same computing platform. The accelerated convergence in the second training stage offsets the cost of the initial alignment phase. Thus, the primary advantage of H-SCALE lies not just in training efficiency, but in its improved final performance. The comparison highlights the impact of H-SCALE on convergence speed, and its higher final performance.

For our experimental results, we begin by plotting the BLEU-4 score on the test set as a function of the number of training steps (Figure 6). In this figure, we compare two variants: Prot2Text-V2 with H-SCALE, and Prot2Text-V2 without contrastive learning (w/o CL). The graph illustrates that Prot2Text-V2 with H-SCALE achieves higher BLEU-4 scores more rapidly, reaching near-maximum performance by training step 4—well ahead of its counterpart. Despite the presence of a variance band, indicated by the shaded region, the H-SCALE model consistently outperforms the other. This demonstrates that aligning the protein encoder output with the semantic space of the pretrained language decoder significantly enhances the model's ability to generate accurate and fluent text, thereby validating the effectiveness of the H-SCALE mechanism.

Compared with the CLIP-based contrastive learning scheme, our H-SCALE approach additionally aligns the standard deviation of sequences of embeddings from two different modalities. Our theoretical motivation is grounded in modeling the sequence of amino-acid token embeddings for a given protein from an underlying multivariate distribution, and the description text token embeddings from another. Under the assumption that these distributions can be reasonably approximated as multivariate Gaussian, they are fully characterized by their first two moments: the mean vector and the covariance matrix. Thus, our approach operationalizes this by aligning the empirical moments of the source (protein) and target (text) distributions. Specifically, we align: (i) The first moment (the mean embedding vector); (ii) A diagonal approximation of the second moment (the element-wise standard deviation vector). This method serves as a computationally efficient proxy for matching the full covariance matrices, effectively aligning the two distributions via the method of moments. Even when the true distributions deviate from a perfect Gaussian, aligning the second moment provides a richer, more robust alignment than matching the mean alone, as it also captures the feature dispersion. While aligning higher-order moments could offer further refinement, our approach represents a principled trade-off between fine-tuning flexibility, computational tractability and alignment fidelity. It provides a lightweight yet effective mechanism for distribution-level, cross-modal bridging. To empirically validate this design, in Table 5 we conduct ablation studies that isolate the contribution of mean and std pooling:

Table 5: Ablation study of Prot2Text-V2 with different alignment strategies. RBT and BBT denote Re-ranking BLEU and BioBERT F1 scores, respectively.

| Ablations | Exct | B-2 | B-4 | R-1 | R-2 | R-L | RBT | BBT |
|---|---|---|---|---|---|---|---|---|
| w/o alignment | 36.52 | 42.70 | 38.98 | 54.71 | 47.62 | 52.90 | 91.33 | 85.58 |
| w/ mean-pooling alignment | 37.01 | 44.39 | 41.23 | 55.30 | 48.27 | 53.62 | 91.61 | 86.19 |
| w/ std-pooling alignment | 33.42 | 37.28 | 33.78 | 50.49 | 43.22 | 48.75 | 90.93 | 84.70 |
| w/ H-SCALE | **39.16** | **46.67** | **43.34** | **57.24** | **50.17** | **55.39** | **91.95** | **86.81** |

# B Computational Complexity

The computational complexity of Prot2Text-V2 arises primarily from its two-stage training process and its multimodal architecture, combining a protein sequence encoder (ESM2-3B), a modality projector, and a decoder-only language model (LLaMA-3.1-8B-Instruct). The ESM2-3B encoder has a standard transformer architecture, therefore, it scales quadratically with the protein sequence length. The decoder-only LLaMA-3.1-8B architecture has 8.2 billion parameters and similarly scales quadratically with the sequence length. The computational complexity of the contrastive learning part is similar, as the model performs forward and backward passes across the encoder and the decoder.

# C Technical Details on SwissProt Dataset

The original human-curated UniProt-SwissProt dataset (Release 2022.02) contains more than 573,230 protein sequences and their corresponding function annotations, likely with many highly similar or redundant sequences, including isoforms, fragments, and repeated entries.

## C.1 CD-HIT Clustering

To ensure robust generalization in protein function generation, created a non-redundant train/validation/test split based on amino acid sequence similarity using CD-HIT, a fast clustering algorithm for biological sequences. CD-HIT groups sequences into non-overlapping clusters based on a user-defined sequence identity threshold. It uses a greedy incremental clustering strategy, selecting the longest sequence as a cluster representative and assigning other sequences to it if they pass the word-count-based similarity threshold, otherwise initializing a new cluster.

In the dataset used for training and evaluation in this study, protein sequences are clustered using CD-HIT to enforce maximum of 40% sequence identity (the minimum CD-HIT identity threshold for protein sequences) between training and validation/test sets. This design choice reflects real-world scenarios where functional models must generalize to novel sequences that differ substantially from those seen during training. The resulting split includes 248,312 proteins for training, 4,172 for validation and 4,203 for testing. We use this established split to allow direct comparison with prior work and to asses our model's ability to transfer knowledge beyond close homologs.

## C.2 Dataset Statistics

During training and inference of Prot2Text-V2, we truncate protein amino acid sequences to 1,024 residues, adhering to the maximum sequence length supported by the pretrained ESM2-3B module. Statistical analysis from Figure 8 of our dataset confirms that the majority of protein sequences fall below this limit, ensuring that truncation has minimal impact on model reliability.

For functional descriptions, we truncate labels to 256 LLaMA-3.1 tokens to balance computational constraints and training efficiency. Since most human-curated UniProt labels are shorter than this threshold as visualized in Figure 8, we omit the end-of-sequence token for truncated cases, encouraging the model to generate longer responses when appropriate. During inference, we permit up to 512 new tokens to enable synthesis of multi-source descriptions, yielding more comprehensive functional predictions.

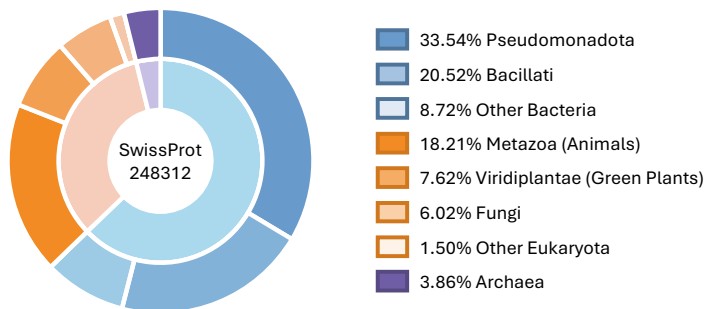

Figure 7: Taxonomic composition of the 248,312 non-redundant protein sequences from SwissProt after CD-HIT clustering. The dataset is dominated by well-studied bacterial groups such as Pseudomonadota (33.5%) and Bacillati (20.5%), which are common in experimental research. It also includes substantial representation from animals (18.2%), green plants (7.6%), and fungi (6.0%), ensuring coverage of key eukaryotic lineages. A smaller fraction comprises Archaea (3.9%) and other taxa, maintaining broad phylogenetic diversity while minimizing redundancy for efficient downstream analysis.

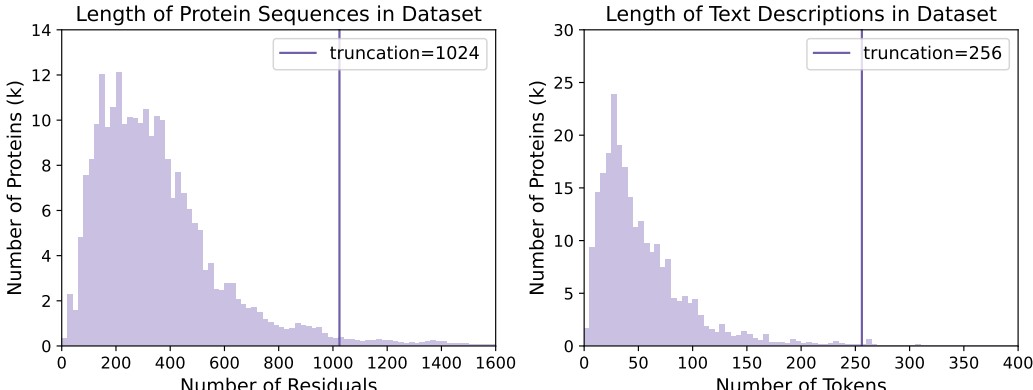

Figure 8: **Left**: Distribution of protein sequence lengths (in residues) in the dataset. Most sequences fall well below the 1,024-residue truncation limit used for ESM2-3B compatibility. **Right**: Distribution of functional description lengths (in tokens). The vertical line marks the 256-token truncation threshold—while most labels are shorter, some require truncation (handled by omitting the EOS token to encourage longer generations).

## C.3 Homology Analysis

The CD-HIT clustering algorithm plays a critical role in our preprocessing pipeline by eliminating redundant protein sequences that could otherwise bias model evaluation. Without such filtering, an overrepresentation of highly similar proteins would artificially inflate performance metrics, as traditional methods can easily achieve high accuracy on strongly homologous sequences. To quantify this effect, we compared two data partitioning strategies: one using CD-HIT-based clustering versus complete random shuffling. This comparison revealed substantial differences in homology distributions between training and test sets. As visually supported by Figure 9, the clustered approach yields more meaningful evaluation conditions by ensuring test proteins maintain sufficient evolutionary distance from training examples. These results confirm that careful sequence clustering is essential for obtaining reliable, generalizable assessments of model performance across diverse protein families.Further details on these metrics can be found in Appendix D.

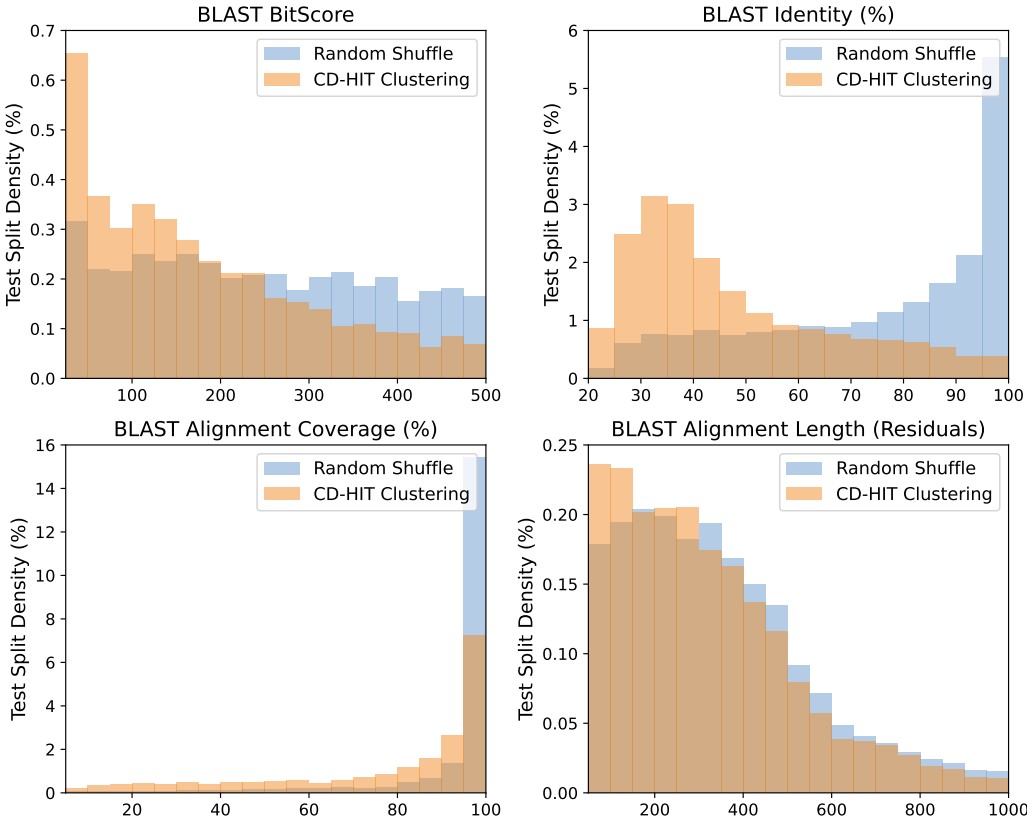

Figure 9: Comparison of homology distributions between CD-HIT clustering and random shuffling approaches. Evolutionary distances between test samples and the training set are quantified using four metrics: BitScore, sequence identity, alignment coverage, and alignment length. The significant leftward shift in distributions demonstrates CD-HIT's effectiveness in rebalancing the dataset toward lower-homology proteins, reducing evaluation bias.

# D  Protein Homology Assessment Using BLAST

To rigorously evaluate the evolutionary distance between test proteins and our training set, we employ the BLAST (Basic Local Alignment Search Tool) algorithm to compute pairwise alignment metrics for each test sequence against all training sequences. For each test protein, we record several key metrics based on its highest-scoring match in the training set: BitScore, Expectation Value (E-Value), Alignment Coverage, Alignment Length, and Sequence Identity.

## D.1  Smith-Waterman Algorithm and Bit Score

The theoretical foundation for sequence alignment scoring is the **Smith-Waterman (S-W) algorithm**, which computes the optimal local alignment between two sequences using dynamic programming. The S-W alignment score $S$ is computed using a substitution matrix (e.g., BLOSUM62) and affine gap penalties. However, the exact S-W algorithm is computationally expensive.

BLAST approximates the S-W score using a fast heuristic, and reports a normalized score called the **bit score**. The bit score $B$ transforms the raw alignment score $S$ into a standardized unit:

$$B = \frac{\lambda S - \ln K}{\ln 2},$$

where:

- $S$ is the raw alignment score,
- $\lambda$ and $K$ are statistical parameters dependent on the scoring matrix and gap penalties,
- $B$ is measured in bits.

The bit score allows for comparison across different BLAST runs and parameter settings, providing a consistent metric for alignment quality.

### D.2  E-value (Expectation Value)

The **E-value** represents the number of alignments with a bit score $\geq B$ expected to occur by chance in a database search. It quantifies the statistical significance of an alignment:

$$E = mn \cdot 2^{-B},$$

where:

- $m$ is the effective length of the query,
- $n$ is the effective length of the database,
- $B$ is the bit score of the alignment.

Lower E-values indicate more statistically significant alignments. An E-value close to 0 suggests the match is unlikely to have occurred by random chance.

### D.3  Alignment Coverage

The **coverage** (or alignment length fraction) measures the fraction of the query sequence that is aligned to the subject sequence. If an alignment spans $L$ residues of a query of length $Q$, then:

$$\text{Coverage} = \frac{L}{Q}.$$

Coverage reflects how much of the query is involved in the alignment and is important for distinguishing between partial and full-length hits.

### D.4  Sequence Identity

The **identity score** quantifies the fraction of aligned positions in which the query and subject residues are exactly the same. If an alignment of length $L$ contains $I$ identical amino acids, the identity score is:

$$\text{Identity} = \frac{I}{L}.$$

Identity provides a simple measure of sequence similarity, but does not account for conservative substitutions, which are captured by the substitution matrix in the bit score.

## E  Uncertainty Estimation of Non-Decomposable Evaluation Metrics

In many machine learning evaluation scenarios—particularly in natural language generation—metrics such as BLEU, ROUGE, and BERTScore are widely used to assess the quality of predictions. Unlike simple metrics such as accuracy or mean squared error, these scores are not decomposable: they depend on aggregate statistics across the entire set of predictions. As a result, computing the metric on individual samples and averaging the results does not yield the same value as computing the metric over the corpus as a whole.

This poses a challenge when attempting to estimate statistical uncertainty, such as constructing confidence intervals. Standard techniques that rely on per-sample variance are invalid in this setting.

Instead, a resampling-based approach over groups of samples must be used. The method presented below estimates the variance and confidence interval of such non-decomposable scores using group-level evaluation, enabling statistically valid uncertainty estimation with minimal computational overhead.

Let $\{x_1, x_2, \ldots, x_{2n}\}$ be a set of $2n$ samples with corresponding system outputs and references. Assume we are able to compute the evaluation metric (e.g., BLEU) only at the group level due to its non-decomposable nature or computational cost.

We partition the $2n$ samples into $k$ disjoint groups, each of size $m = \frac{2n}{k}$. Let the evaluation metric computed over group $i$ be denoted by:

$$s_i = \text{Metric}(\text{Group}_i), \quad \text{for } i = 1, 2, \ldots, k.$$

Let the average of the group-level scores be:

$$\bar{s} = \frac{1}{k} \sum_{i=1}^{k} s_i.$$

The sample variance of the metric across groups serves as an estimate of the uncertainty in the overall evaluation:

$$\hat{\sigma}^2 = \frac{1}{k-1} \sum_{i=1}^{k} (s_i - \bar{s})^2.$$

We use the grand mean $\bar{x}$ as a point estimate of the population mean $\mu$:

$$\hat{\mu} = \bar{x}.$$

Assuming approximate normality of the group-mean estimator (justified by the Central Limit Theorem for sufficiently large $k$), a 2-sigma confidence interval for the metric is given by:

$$\left[ \bar{s} - 2 \cdot \hat{\sigma}, \ \bar{s} + 2 \cdot \hat{\sigma} \right].$$

This interval quantifies the uncertainty of the evaluation score under random resampling of the data and accounts for the non-decomposability of the metric.

# F  Protein Function Prediction with Additional Text Fields

During training, we allow users to optionally include two additional text fields: the protein name and its taxonomy. This enables the model to support generation in cases where such information is available. To accommodate this, we incorporate these fields into the training process with an 80% dropout rate, ensuring the model remains robust in this scenario. In our main text although this training is conducted our test set had 100% dropout to guarantee that we will remain fair in all the previously presented scenarios.

In Table 6 and Figure 10, we present additional experiments in which the protein name is included. We hypothesize that protein names offer valuable semantic cues that can improve functional prediction, especially when aligned with the LLaMA decoder's latent representations. In Table 6, we evaluate Prot2Text-V2+ (w/ H-SCALE)—where the '+' indicates inclusion of the protein name—against several general-purpose LLMs: LLaMA 3.1, Claude 3.5 Sonnet, and GPT-4o-mini. Each of these models is prompted with both the amino acid sequence and protein name, under zero-shot and five-shot settings, without access to external retrieval. As shown in Figure 10, the inclusion of these additional fields can significantly enhance function prediction performance. Nevertheless, in a semantic context, the original Prot2Text-V2 metrics remain comparable, demonstrating competitive performance even without the added fields.

It is possible to see in Table 6 that even in the scenario in which these extra fields are available our model surpasses all models among all of the presented metrics.

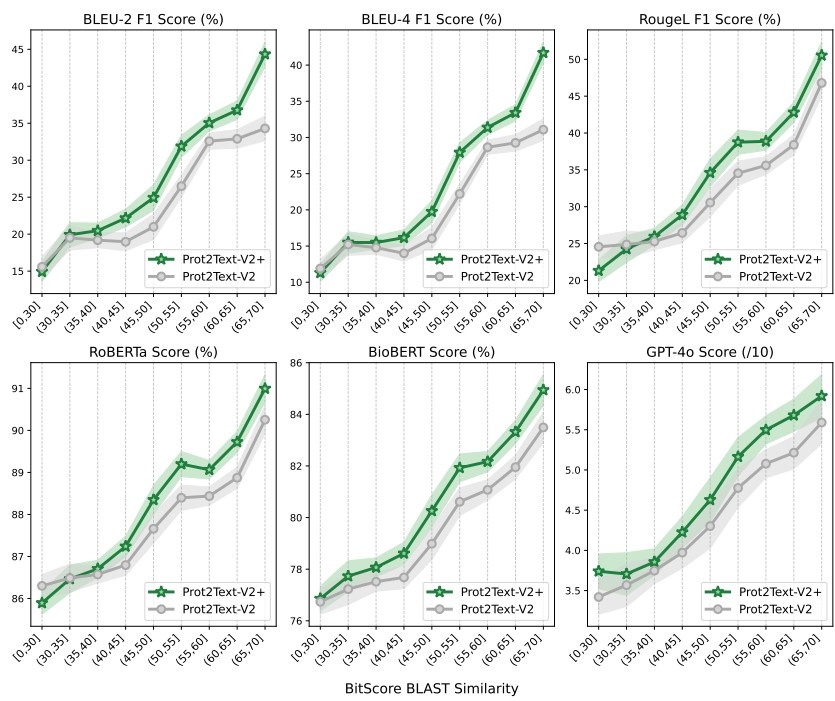

Figure 10: Evaluation of protein function descriptions across BLAST sequence bitscore bins (%). Prot2Text-V2+, which includes the protein name as input, achieves the best performance.

Table 6: Comparative evaluation of Prot2Text-V2+ (w/ H-SCALE), which incorporates the protein name as input, against general-purpose LLMs including LLaMA 3.1, Claude 3.5 Sonnet, and GPT-4o-mini. Results highlight the advantage of domain-specific modeling when protein context is explicitly provided.

| Model | Exct | B-2 | B-4 | R-1 | R-2 | R-L | RBT | BBT | GPT |
|---|---|---|---|---|---|---|---|---|---|
| LLaMA-3.1 (0-shot) | - | 7.63 | 2.80 | 23.83 | 7.76 | 18.89 | 85.84 | 77.22 | 45.03 |
| LLaMA-3.1 (5-shot) | - | 7.99 | 3.18 | 25.60 | 8.97 | 20.55 | 86.40 | 77.74 | 44.71 |
| Claude-3.5-Sonnet (0-shot) | - | 9.78 | 4.22 | 25.98 | 9.28 | 20.04 | 86.02 | 79.23 | - |
| Claude-3.5-Sonnet (1-shot) | - | 15.72 | 7.79 | 28.24 | 11.76 | 21.70 | 86.56 | 79.71 | - |
| GPT-4o-mini (0-shot) | - | 4.79 | 1.18 | 14.86 | 2.79 | 10.32 | 82.69 | 73.46 | - |
| GPT-4o-mini (1-shot) | - | 8.53 | 2.46 | 19.04 | 4.39 | 14.41 | 84.24 | 73.96 | - |
| Prot2Text-V2 (w/ H-SCALE) | 39.16 | 46.67 | 43.34 | 57.24 | 50.17 | 55.39 | 91.95 | 86.81 | 64.58 |
| Prot2Text-V2+ (w/ H-SCALE) | **43.85** | **55.64** | **52.46** | **63.46** | **57.03** | **61.70** | **93.17** | **89.01** | **71.29** |

# G Qualitative Evaluation

Table 7 presents qualitative examples of predictions generated by Prot2Text-V2. The table includes the original UniProt description for each selected protein, serving as the reference label. Alongside this, we show the corresponding prediction produced by Prot2Text-V2, the score assigned by the LLM-as-a-judge evaluation framework, and the explanation provided for that score.

We selected seven proteins from the target group characterized by low BLAST bitscore, specifically within the 0–50 range. In this challenging weak-homology setting, our model generally captures the core concepts conveyed in the original UniProt descriptions, which serve as the reference labels. While the predictions may not replicate every detail, they often reflect the correct functional context. For samples receiving lower LLM-as-a-judge scores (e.g., 5 or 6), we observe occasional issues with over-specification or the introduction of overly confident assertions, likely due to the model's attempt to compensate for limited similarity-based cues. On the other hand, predictions with higher judge scores tend to miss only minor detail information mentioned in the reference, but omitted in the output, while still maintaining accurate and coherent functional descriptions.

From the qualitative results, we observe that Prot2Text-V2 is capable of generating function descriptions that align with the referenced UniProt annotations, even in challenging weak-homology scenarios (BLAST BitScore under 50). This suggests that the model is not simply memorizing known examples or relying on close sequence neighbors, but is instead learning to generalize functional patterns from the input modalities.

Table 7: Qualitative examples of protein function predictions generated by Prot2Text-V2 for proteins with low BLAST identity scores (0–30%). We show the reference UniProt description, the model's prediction, the explanation for what is considered wrong and right in the prediction, and the LLM-as-a-judge score for it based on the explanation. These examples illustrate the model's ability to generalize functional information even in low-homology settings.

| Protein | UniProt Description | Prot2Text_V2 Description | Judge Explanation | GPT Score |
|---|---|---|---|---|
| LPP20_HELPJ | Could play a role in the pathogenesis of H. pylori by serving as an inflammatory mediator. | May be involved in virulence. | The prediction captures a general function related to virulence, but lacks specificity on inflammatory mediation and omits H. pylori. | 5 |
| FLGB_BACSU | Structural component of flagellum; part of the rod structure of the flagellar basal body. | Describes the rod as a central, helical, flexible structure in the basal body transmitting torque. | Captures general relevance to flagellar apparatus, but omits specific motility role and lacks clarity on structural function. | 6 |
| TAS3_SCHPO | Part of the RNAi pathway important for heterochromatin formation and chromosome segregation; member of the RITS complex. | Involved in RNAi pathway as a transcriptional repressor; describes dsRNA generation and interaction with RNase H1 or Dicer-like enzymes. | Touches on RNAi and dsRNA roles, but inaccurately attributes siRNA source and omits key roles in heterochromatin and chromosome dynamics. | 6 |
| ATOX1_BACCR | Immunity component of an LXG toxin-immunity module; neutralizes RNase activity of BC_0920. | Describes LXG module involvement, kin selection, biofilm competition; neutralizes BC_0922 toxin in E. coli. | Captures immunity concept well, but introduces unrelated toxin (BC_0922) and extra context not in original description. | 6 |
| RSIX_BACSU | Anti-sigma factor for ECF sigma factor SigX; inhibits and stabilizes SigX. | Negative regulator of sigma-X; impacts xerCD operon transcription and protein production. | Describes inhibition well but omits ECF context and stabilization; partial view of functional implications. | 7 |
| CPT_STRVP | Inactivates chloramphenicol by transferring gamma-phosphate of ATP to C-3' hydroxyl group. | Phosphorylates 3-hydroxyl group of phenolic ring to inactivate chloramphenicol. | Matches key function but less specific about ATP and hydroxyl group position; mostly accurate. | 8 |
| PAAA_ECOLI | Catalytic subunit of enzyme system reducing PA-CoA to form epoxyphenylacetyl-CoA; incorporates one oxygen atom. | Monooxygenase in same system; catalyzes epoxidation; can use alternative substrates; prefers NADPH. | Highly accurate and informative; slight wording variation from ground truth, otherwise excellent. | 9 |

## G.1 Human Expert Evaluation

To further validate our predictions, we compared our model's protein function predictions to those of eight widely used and referenced large language models (LLMs), all based solely on sequence input without internet access. To assess the quality of these predictions, we asked two biology experts to independently and blindly evaluate each model's predictions for nine well-known proteins in the field.

**Score scale:** Throughout this qualitative verification we adopted an evaluation scale from 1 to 5 and asked them to evaluate the correctness and completeness of these generated descriptions based on their domain knowledge and familiarity with these protein sequences. The rating of each model's output should be done according to how well it could reflect the known or probable function of the respective protein. We further describe our scale as:

- Score = 1 for completely inaccurate generations. The generated description is biologically implausible or entirely unrelated to the known function of the protein.

- Score = 2 for poorly accurate generations. The description contains significant inaccuracies or misinterpretations, with little relevance to the known function.

- Score = 3 for moderately accurate generations. The description is partially correct but includes vague, generic, or somewhat misleading elements.

- Score = 4 for mostly accurate generations. The output is largely consistent with the known function, with only minor errors or omissions.

- Score = 5 for highly accurate generations. The description is biologically sound, detailed, and aligns closely with the known or expected function of the protein.

**LLM models evaluated:** In this qualitative assessment, we evaluate a selection of widely used, general-purpose generative LLMs: GPT-4, Grok, Claude 3.7 Sonnet, Gemini 2, LLaMA 4, and Mistral 7B. These models were chosen due to their multi-domain capabilities, and their capacity to generate coherent, context-aware responses from textual input. Although not explicitly trained in protein-specific data, such general-purpose LLMs have demonstrated aptitude in biological tasks because of their exposure to large-scale textual corpora, including protein databases with stored descriptions. Evaluating them in this context allows us to assess their zero-shot generalization ability and measure how well language-driven reasoning alone can approximate biological insight.

To contextualize the performance of our proposed model, Prot2Text-V2, we also include comparisons with prior protein function generation approaches: Prot2Text, which introduced protein-to-text generation using encoder-decoder architectures; and Evolla-10B, a domain-specific model trained on a dataset approximately 200 times larger than the one we currently use, designed for structured biological knowledge decoding. While we do not aim to directly compete with a model trained on a dataset much larger then ours, like Evolla-10B, including them in our evaluation provides valuable insight into the effectiveness and efficiency of Prot2Text-V2 under significantly more constrained training resources, highlighting its potential in low-resource or scalable deployment scenarios. This comparative framework enables us to benchmark Prot2Text-V2 not only against domain-specific baselines but also against the best general-purpose reasoning engines currently available.

**Human expert assessment:** According to the experts, some of the models provided highly accurate and detailed descriptions, receiving top scores (4 or 5). However, several others were either not relevant to the protein in question or included explanations that were unclear and difficult to follow. Notably, none of the submissions addressed species-specific differences, which is understandable given the high degree of amino acid conservation across species from humans to dog, cow, yeast, etc. It is worth noting that, according to them, the degree of detail the LLMs returned was strikingly different, and in some cases, the models generated extensive information on residues phosphorylated, binding partners and other molecular/cellular details. It was with such details that the assessors spent most time as there were instances of incorrect information that created concern on overall output validity and trustworthiness.

**Results and discussion:** The resulting expert ratings are presented in Table 8. The blinded biological assessment of LLM outputs showed good correlation between the two assessors. Based on the average scores, our model, Prot2Text-V2, clearly outperforms most general-purpose LLMs, reinforcing the value of domain-specific fine-tuning even when using a relatively small dataset. Although Evolla-10B achieves higher scores, as expected from a large-scale model trained on a dataset with 200 times more proteins than what we originally trained our model with, this comparison should be interpreted with caution. Evolla-10B and Claude 3.7 benefit from access to an extensive training set, making it likely that evaluated proteins or closely related ones were seen during training. They are as well capable of predicting, from their extensive training database, the names of the protein sequences we use, which facilitates in generating the prediction. In contrast, Prot2Text-V2

Table 8: Human annotations from biology experts (A and B) for nine highly understood proteins across selected models. The score scales are defined as: (1) Completely inaccurate; (2) Poor accuracy; (3) Moderately accurate; (4) Mostly accurate; and (5) Highly accurate. The models used in this comparison were: BLAST (BST); OpenAI-GPT-4o (GPT4); Grok-3 (GRK); Claude-3.7-Sonnet (CLD3); Google-Gemini-2 (GM2); Meta-LLaMA-4 (LM), Mistral-7B (MST); Prot2Text (PTV); our Prot2Text-V2 (PTV2); ProtT3 (PTT3); and Evolla-10B (EVL)

| Protein | BST | | GPT4 | | GRK | | CLD3 | | GM2 | | LM | | MST | | PTV | | PTV2 | | PTT3 | | EVL | |
|---|---|---|---|---|---|---|---|---|---|---|---|---|---|---|---|---|---|---|---|---|---|---|
| Evaluator | A | B | A | B | A | B | A | B | A | B | A | B | A | B | A | B | A | B | A | B | A | B |
| Q9H2D6·TARA_H | 4 | 5 | 1 | 1 | 2 | 3 | 2 | 2 | 2 | 1 | 1 | 1 | 3 | 2 | 1 | 1 | 5 | 5 | 1 | 1 | 1 | 1 |
| B2RTY4·MYO9A_H | 2 | 2 | 1 | 1 | 3 | 4 | 4 | 4 | 1 | 1 | 3 | 3 | 2 | 1 | 2 | 3 | 4 | 4 | 3 | 3 | 5 | 3 |
| Q29537·P53_C | 5 | 3 | 5 | 5 | 3 | 4 | 5 | 5 | 4 | 3 | 2 | 3 | 2 | 3 | 5 | 4 | 2 | 3 | 4 | 3 | 5 | 5 |
| Q32KY4·CDK4_B | 5 | 4 | 2 | 3 | 3 | 3 | 4 | 4 | 4 | 3 | 1 | 2 | 1 | 1 | 4 | 5 | 1 | 3 | 4 | 5 | 5 | 4 |
| P0CY46·EGFR_A | 1 | 1 | 5 | 5 | 4 | 3 | 5 | 5 | 4 | 4 | 3 | 5 | 1 | 3 | 2 | 1 | 1 | 1 | 2 | 2 | 5 | 3 |
| P04806·HXKA_Y | 5 | 5 | 1 | 1 | 2 | 2 | 4 | 5 | 1 | 1 | 2 | 1 | 3 | 1 | 4 | 5 | 4 | 5 | 4 | 5 | 5 | 5 |
| O14929·HAT1_H | 2 | 3 | 1 | 1 | 2 | 2 | 3 | 3 | 1 | 2 | 2 | 1 | 2 | 2 | 1 | 2 | 1 | 2 | 2 | 5 | 2 | 1 |
| P00846·ATP6_H | 4 | 5 | 1 | 2 | 2 | 2 | 5 | 5 | 4 | 2 | 3 | 2 | 5 | 4 | 5 | 4 | 4 | 4 | 4 | 5 | 5 | 5 |
| P06213·INSR_H | 1 | 1 | 5 | 5 | 3 | 5 | 2 | 4 | 5 | 5 | 1 | 2 | 4 | 4 | 4 | 4 | 4 | 4 | 1 | 1 | 3 | 3 |
| **Avg Scores per model** | 3.22 | | 2.55 | | 2.88 | | 3.77 | | 2.55 | | 2.60 | | 1.94 | | 3.38 | | 3.61 | | 3.06 | | 4.21 | |

Table 9: Pearson correlation coefficient between automatic evaluation metrics and human opinions.

| Metric | B-2 | B-4 | R-L | RBT | BBT | GPT |
|---|---|---|---|---|---|---|
| **Human** | 0.63 | 0.61 | 0.59 | 0.55 | 0.68 | **0.73** |

was trained on a much smaller and more curated dataset, and yet achieves scores within one point of Evolla. This demonstrates the efficiency and generalization capability of our model, suggesting that with modest scaling, Prot2Text-V2 has the potential to match or even surpass the performance of significantly larger and more resource-intensive models. We also computed, in Table 9, the Pearson correlation of the metrics from Table 1 with the expert evaluations. The Pearson correlations with expert evaluations show that while BLEU and ROUGE have moderate alignment ( 0.60), BioBERT (0.68) and GPT score (0.73) correlate more strongly, likely due to their deeper semantic understanding in the biological domain.

# H   LLM as a Judge

Although there are studies that use LLMs as judges in experiments, to our knowledge, no definitive or universally accepted protocol has been established for this approach. In many cases, research that employs LLMs as evaluators does not clearly specify the prompts or design choices used. This lack of a standardized protocol poses several challenges, including issues with reproducibility, transparency, the ability to compare models, and ensuring the validity of evaluations. We describe our protocol below.

The configurations for this experiment involved using GPT-4-mini as the LLM judge. In addition, to minimize variability in the responses, we set the temperature to 0, making the output deterministic. This ensures that the model produces the same result every time for the same input, reducing the possibility of diverse or outlier predictions that could introduce noise into the evaluation. Finally, we also developed a prompt to compare our predictions for protein function and their respective labels, as shown bellow:

> **Prompt:**
> I will provide you with three pieces of text:
>
> - **Protein name:** This is the name of the protein.
> - **Ground Truth Description:** This is the curated, authoritative description of a protein's function (sourced from UniProt).
> - **Predicted Description:** This is an automated prediction of the protein's function.
>
> **Task:**
>
> Evaluate how well the predicted description matches the ground truth description. Your evaluation
> should focus on:
>
> - **Key Functional Terms:** Check if the prediction includes the essential keywords and concepts present in the ground truth.
> - **Specificity:** Assess whether the prediction captures the specific details (e.g., catalytic activity, binding sites, cellular localization) that are mentioned in the ground truth
> - **Completeness**: Determine if the prediction covers all major aspects of the protein function described in the ground truth.
> - **Accuracy:** Evaluate how accurately the prediction reflects the true function as described, including any relevant mechanisms or contextual details.
>
> **Scoring Guidelines:**
>
> Provide a final score on a scale from 0 to 10, where:
>
> - **0:** No similarity or relevant matching elements between the prediction and the ground truth.
> - **10:** A perfect match with all key elements of the ground truth correctly captured.
>
> **Output Format:**
>
> Your output must include:
> A single line with the score, formatted as:
>
> **Score:** X (where X is an integer between 0 and 10) Another line with a brief explanation summarizing the reasoning behind the score.
> **Explanation:**This explanation should highlight the strengths and weaknesses of the prediction relative to the ground truth.

Figure 11: OpenAI GPT-4o-mini user prompt instructions for LLM-as-a-judge evaluation. Name, labeled description and prediction description of every protein are be provided afterwards.

This experiment aimed to evaluate the predictions of Blast, Prot2Text, and the variations of Prot2Text-V2. The results for various Blast similarity scores are shown in Figure 5. From this, we can see that all proposed variations outperform Blast in low similarity scenarios.

# I   Safeguards

Prot2Text-V2 demonstrates strong potential for protein function prediction, though we cannot rule out the possibility of occasional inaccuracies, particularly with novel or poorly characterized proteins, so results should be interpreted with care and verified where possible. While the model is designed for beneficial research applications, we encourage its use in responsible and well-regulated settings to ensure alignment with ethical and safety standards.

