# OpenReview forum: "Prot2Text-V2: Protein Function Prediction with Multimodal Contrastive Alignment"
_NeurIPS.cc/2025/Conference — NeurIPS 2025 poster_

### Official Review · Reviewer_qp8d · 2025-06-26

**Clarity:** 3
**Significance:** 2
**Originality:** 3
**Rating:** 4
**Confidence:** 5

**Summary:**

This paper presents Prot2Text-V2, a multimodal sequence-to-text model for generating free-form protein function descriptions from amino acid sequences. It integrates a protein encoder (ESM2) with a decoder (LLaMA) via a lightweight projector and introduces a contrastive alignment strategy to improve semantic consistency. The method is evaluated under low-homology settings and shows strong performance compared to prior baselines.

**Questions:**

The appendix shows that including protein names significantly boosts performance. Could the authors clarify to what extent the model relies on such semantic cues, and whether this impacts its generalization to unnamed or novel proteins?

**Ethical Concerns:**

["NO or VERY MINOR ethics concerns only"]

**Final Justification:**

Thank you for the new results—they address a major concern and clearly show improved low-homology performance over ProtT3. I was originally referring to comparisons with other methods specifically designed for low-homology settings, which are not discussed. Also, the performance gap between low and high homology bins raises questions about the impact of training data distribution. Finally, I appreciate the clarifications and additional effort.

**Limitations:**

The authors clearly acknowledge the lack of wet-lab validation and discuss potential extensions to structural modalities and reasoning capabilities.

**Paper Formatting Concerns:**

No major formatting issues observed.

**Quality:**

3

**Strengths And Weaknesses:**

Strength:
The paper addresses the underexplored task of generating free-text protein function descriptions directly from sequences, which has practical value in annotating novel proteins beyond fixed ontology terms.

Weaknesses
1. The core modeling choices, including the use of a nonlinear projector and decoder-only architecture, are more of engineering simplifications than methodological innovations. While effective, these components largely build on existing paradigms rather than introducing fundamentally new ideas.

2. Although the paper emphasizes generalization under weak-homology settings, it lacks direct comparison to other function prediction methods that also claim to improve generalization. In addition, ProtT3 is only evaluated under random splits and not under the same low-homology CD-HIT split, limiting the fairness of the comparison.

---

> ### Author Rebuttal · Authors · 2025-07-31
>
> Dear reviewer:
>
> Thank you for your thoughtful review and for recognizing the value of our work in annotating novel proteins. We appreciate your feedback, which points to important areas where we can further clarify our contributions and strengthen our experimental comparisons. We address the concerns below:
>
> **[Weakness 1 (W1) on Novelty]**
>
> While we build upon established components like ESM and LLMs, we believe that our proposed model Prot2Text-V2 has significant and novel methodological improvements over existing multimodal LLM designs. While there are some similarities with ProtT3 architecture, our model departs in the following essential respects:
>
> ProtT3 relies on a Q-Former, an additional attention module with learnable queries of a fixed number and a redundant text encoder, to align the protein and text modalities. In contrast, Prot2Text-V2 follows a decoder-only design with a lightweight projector and with a parameter-free alignment method.  Rather than learning alignment implicitly through a query-attention module, H-SCALE directly aligns the projected protein representations to intermediate text embeddings with a new contrastive learning objective. This eliminates the learned bottleneck, preserves the full expressivity of the pretrained protein encoder, and avoids added parameters, latency, and memory, and also allows us to use a more powerful LLM decoder.  We also want to emphasize that contrastive learning from intermediate LLM embeddings is not trivial, and it has not been done in any prior work, since all previous methods use an additional text encoder.
>
> Compared with the CLIP-based contrastive learning scheme, our H-SCALE approach additionally aligns the standard deviation of sequences of embeddings from two different modalities. Our theoretical motivation is grounded in modeling the sequence of amino-acid token embeddings for a given protein from an underlying multivariate distribution, and the description text token embeddings from another. Under the assumption that these distributions can be reasonably approximated as multivariate Gaussian, they are fully characterized by their first two moments: the mean vector and the covariance matrix. Thus, our approach aligns the empirical moments of the source (protein) and target (text) distributions.
>
> **[W2 on Comparison with ProtT3]**
>
> We fully agree that evaluating ProtT3 under the same low-homology settings is important for a fair and thorough comparison. In the tables below, we filtered test proteins with the same bitscore ranges as shown in Figure 5, and made a direct comparison between different approaches on the same ranges. Additional evaluation results on BLEU-4 F1 score and BioBert F1 score are shown below.
>
> Table 2: BLEU-4 F1 Score of ProtT3 across BLAST sequence bitscore bins (%)
>
> | Model\BitScore | [0,30] | (30,35] | (35,40] | (40,45] | (45,50] | (50,55] | (55,60] | (60,65] | (65,70] |
> |---|---|---|---|---|---|---|---|---|---|
> | Prot2Text | 6.58 | 6.41 | 7.51 | 8.63 | 12.58 | 16.03 | 20.76 | 21.68 | 27.04 |
> | ProtT3 | 6.19 | 5.26 | 7.12 | 9.76 | 13.10 | 10.87 | 18.97 | **30.95** | **40.82** |
> | Prot2Text-V2 | **11.89** | **15.21** | **14.78** | **14.00** | **16.05** | **22.20** | **28.65** | 29.26 | 31.08 |
>
> Table 3: BioBert F1 Score of ProtT3 across BLAST sequence bitscore bins (%)
>
> | Model\BitScore | [0,30] | (30,35] | (35,40] | (40,45] | (45,50] | (50,55] | (55,60] | (60,65] | (65,70] |
> |---|---|---|---|---|---|---|---|---|---|
> | Prot2Text | 73.38 | 73.92 | 74.30 | 74.11 | 75.04 | 76.64 | 78.24 | 78.31 | 78.98 |
> | ProtT3 | 67.57 | 66.44 | 69.19 | 74.37 | **79.73** | 78.11 | 76.44 | 81.06 | **83.99** |
> | Prot2Text-V2 | **76.74** | **77.23** | **77.52** | **77.68** | 78.99 | **80.61** | **81.07** | **81.95** | 83.49 |
>
> These two additional tables further prove that Prot2Text-V2 outperforms both Prot2Text and ProtT3 across nearly all low-homology bins, especially in the [0,40%] range in both BLEU-4 F1 score and BioBERT F1 score.
>
> However, we restate that the original train-test split of the ProtT3 paper is strongly imbalanced and lacks attention to the low-homology proteins. Although our test set contains 60% fewer proteins overall, it includes more than twice as many low-homology proteins compared to theirs.
>
> **[Question on Protein Name]**
>
> In our main experiments, protein names are not included in the training or in the inference, ensuring that the model does not trivially memorize or rely on name identifiers. The analysis in the appendix shows that including names boosts performance, as expected, but this setting was only used to probe the upper bound of the model’s capacity, not in our default evaluation. Furthermore, our low-homology setup ensures that the model generalizes to novel and unseen proteins, without the name information.

---

> > ### Comment · Reviewer_qp8d · 2025-08-03
> >
> > Thank you for the new results—they address a major concern and clearly show improved low-homology performance over ProtT3. I was originally referring to comparisons with other methods specifically designed for low-homology settings, which are not discussed. Also, the performance gap between low and high homology bins raises questions about the impact of training data distribution. Finally, I appreciate the clarifications and additional effort.

---

> ### Author Response · Authors · 2025-08-04
> **Response to Reviewer qp8d**
>
> Dear reviewer:
>
> Thank you very much for your prompt response. We are truly grateful that you find our additional experimental results address your major concerns.
>
> **(1)** Regarding other methods specifically targeting the low-homology domain, we conducted a thorough review of the literature and found that relatively few approaches address this aspect of the problem, as we noted in our manuscript. A recent study by Juntong Wu et al. [1] demonstrated that many existing protein-related models do not outperform retrieval-based approaches (most notably the BLAST predictor which we included in our comparisons) primarily due to data leakage,which also contributes to their limited effectiveness in remote-homology settings. The Prot2Text method [2] attempted to mitigate such concern by introducing a new clustering-based partition strategy, and our approach outperformed it due to our novel alignment strategies and many other architectural innovations. While Zuobai Zhang et al. [3] discussed the influence of sequence similarity on protein encoders, their work was focused on structured GO label annotation rather than free-text generation. This limits the expressiveness of the annotations and leaves the comparison irrealistic. Overall, we have compared our method against most relevant and recent approaches in this area, and it consistently achieves superior results.
>
> **(2)** We appreciate your comments about the performance gap across different homology domains. In our view, this gap isn’t mainly due to an imbalance in the training data, but rather reflects the real challenge of generalizing to low-homology proteins. It’s a well-known principle in molecular biology and bioinformatics that similar protein sequences often have similar functions. This idea forms the basis of both manual annotations and retrieval-based prediction methods. As such, it is expected that our model does better on test proteins that are more similar to those in the training set, since they align more closely with what the model has learned. Meanwhile, in order to reduce bias, we clustered the training data to remove redundancy while keeping a broad range of sequences. However, we respectfully clarify that the homology-based split of the test set was done only for evaluation purposes. If we had aimed for a more homology-balanced test set, many samples would have been highly similar to training examples. That could have inflated the results and made it harder to measure how well the model handles novel cases. Still, we acknowledge that some bias in homology coverage might remain, and we’re happy to look into this more in future work.
>
> Once again, we sincerely thank you for your positive review and constructive feedback. Should you have any further questions or suggestions, we would be glad to address them.
>
> ---
>
> [1] Juntong Wu et al., Rethinking Text-based Protein Understanding: Retrieval or LLM? arXiv 2505.20354
>
> [2] Hadi Abdine et al., Prot2Text: Multimodal Protein's Function Generation with GNNs and Transformers. AAAI 2024.
>
> [3] Zuobai Zhang et al., Structure-Informed Protein Language Model. GEM workshop, ICLR 2024.

---

### Official Review · Reviewer_4fdJ · 2025-07-02

**Clarity:** 3
**Significance:** 3
**Originality:** 3
**Rating:** 4
**Confidence:** 3

**Summary:**

The paper introduces Prot2Text-V2, a model that aligns protein embeddings with an LLM's feature space using a contrastive strategy, H-SCALE. Following this alignment, the model fine-tunes a LLaMA-3.1-8B-Instruct decoder with LoRA to generate functional protein descriptions. Experiments on a low-homology dataset show the effectiveness of the proposed method.

**Questions:**

See the Weakness part

**Ethical Concerns:**

["NO or VERY MINOR ethics concerns only"]

**Final Justification:**

Thank you, authors, for providing a detailed rebuttal. The experimental results presented by the authors are convincing, for example, the tables in W1.3, W2, and W3. Regarding novelty, I acknowledge the authors' claim that their method differs from previous approaches. However, it seems more like a combination of several existing techniques. I'm not underestimating the contribution of this work, but I think the novelty is still relatively limited. Taking everything into account, since the authors have provided more experimental results, including comparisons to ProtT3 on low-homology and P2T-GPT, I would prefer to give a result that is somewhere between a weak reject and a weak accept, leaning towards a weak accept.

**Limitations:**

yes

**Quality:**

2

**Strengths And Weaknesses:**

Strengths
1.	The paper solves a meaningful problem, protein-to-text generation.
2.	This paper is easy to understand.
3.	The paper uses a contrastive strategy to better align protein representations with an LLM's feature space, which enhances the model's capability for protein function prediction.

Weaknesses
1.	The novelty is incremental. Most components in this work have existed before, such as ESM, finetuning the LLM with Lora, and contrastive learning. The main contribution of this paper lies in H-SCALE. However, the paper fails to provide a clear theoretical motivation for why aligning both mean and standard deviation is the optimal strategy. Furthermore, it lacks the necessary experimental ablation study (e.g., comparing mean-only, std-only, and the hybrid approaches) to validate this specific design.
2.	The baseline comparisons could be further strengthened, particularly with respect to the state-of-the-art method, ProtT3. Although the paper includes an evaluation against ProtT3, this is conducted on ProtT3's original dataset, which follows a random split. However, one of the core contributions in this work emphasizes generalization under low-homology settings. Therefore, a direct and fair comparison on the more challenging low-homology dataset would be essential to convincingly demonstrate state-of-the-art generalization performance.
3.	P2T-GPT [1] also conducted experiments on proteins from a similar database (SwissProt). It trains a protein encoder and GPT from scratch and demonstrates promising results. It would be better to include an analysis and comparison with P2T-GPT.
[1] Protein Captioning: Bridging the Gap between Protein Sequences and Natural Languages. In TOMM, 2024

---

> ### Author Rebuttal · Authors · 2025-07-31
>
> Dear reviewer:
>
> Thank you for your valuable feedback. We are very grateful for your recognizing the value of the protein captioning problem, the clarity of the paper and the improved performance.
>
> **[Weakness-1.1 (W1.1) on Novelty]**
>
> While we build upon established components like ESM and LLMs, we believe that our proposed model Prot2Text-V2 has significant and novel methodological improvements over existing multimodal LLM designs. While there are some similarities with ProtT3 architecture, our model departs in the following essential respects:
> ProtT3 relies on a Q-Former, an additional attention module with learnable queries of a fixed number and a redundant text encoder, to align the protein and text modalities. In contrast, Prot2Text-V2 follows a decoder-only design with a lightweight projector and with a parameter-free alignment method.  Rather than learning alignment implicitly through a query-attention module, H-SCALE directly aligns the projected protein representations to intermediate text embeddings with a new contrastive learning objective. This eliminates the learned bottleneck, preserves the full expressivity of the pretrained protein encoder, and avoids added parameters, latency, and memory, and also allows us to use a more powerful LLM decoder. We also want to emphasize that contrastive learning from intermediate LLM embeddings is not trivial, and it has not been done in any prior work, since all previous methods use an additional text encoder.
>
> **[W1.2 on Theoretical Motivation]**
>
> Compared with the CLIP-based contrastive learning scheme, our H-SCALE approach additionally aligns the standard deviation of sequences of embeddings from two different modalities. Our theoretical motivation is grounded in modeling the sequence of amino-acid token embeddings for a given protein from an underlying multivariate distribution, and the description text token embeddings from another. Under the assumption that these distributions can be reasonably approximated as multivariate Gaussian, they are fully characterized by their first two moments: the mean vector and the covariance matrix. Thus, our approach operationalizes this by aligning the empirical moments of the source (protein) and target (text) distributions. Specifically, we align:
>
> 1. The first moment (the mean embedding vector).
>
> 2. A diagonal approximation of the second moment (the element-wise standard deviation vector).
>
> This method serves as a computationally efficient proxy for matching the full covariance matrices, effectively aligning the two distributions via the method of moments. Even when the true distributions deviate from a perfect Gaussian, aligning the second moment provides a richer, more robust alignment than matching the mean alone, as it also captures the feature dispersion. While aligning higher-order moments could offer further refinement, our approach represents a principled trade-off between fine-tuning flexibility, computational tractability and alignment fidelity. It provides a lightweight yet effective mechanism for distribution-level, cross-modal bridging.
>
> **[W1.3 on Ablation Study for CL]**
>
> To empirically validate this design, we conduct ablation studies that isolate the contribution of mean and std pooling:
>
> Table 1: Ablation study of Prot2Text-V2 with Different Contrastive Learning Schemes
>
> | **Ablations** | **Exct** | **B-2** | **B-4** | **R-1** | **R-2** | **R-L** | **RBT** | **BBT** |
> |---|---|---|---|---|---|---|---|---|
> | w/o alignment | 36.52 | 42.70 | 38.98 | 54.71 | 47.62 | 52.90 | 91.33 | 85.58 |
> | w/ mean-pooling alignment | 37.01 | 44.39 | 41.23 | 55.30 | 48.27 | 53.62 | 91.61 | 86.19 |
> | w/ std-pooling alignment | 33.42 | 37.28 | 33.78 | 50.49 | 43.22 | 48.75 | 90.93 | 84.70 |
> | w/ H-SCALE | 39.16 | 46.67 | 43.34 | 57.24 | 50.17 | 55.39 | 91.95 | 86.81 |
>
> These results confirm that aligning both mean and variance yields the strongest performance, empirically supporting our design. The results also suggest that aligning only the second-order moment is not sufficient, which supports our decision to align both first- and second-order moments simultaneously for better performance. We sincerely thank the reviewer for the insightful suggestion, and we will include these findings and the corresponding analysis in the camera-ready version of the paper.
>
> **[W2 on Remote Homolog Analysis for ProtT3]**
>
> We fully agree that evaluating ProtT3 under the same low-homology settings is important for a fair and thorough comparison. In the tables below, we filtered test proteins with the same bitscore ranges as shown in Figure 5, and made a direct comparison between different approaches on the same ranges. Additional evaluation results on BLEU-4 F1 score and BioBert F1 score are shown below.
>
> Table 2: BLEU-4 F1 Score of ProtT3 across BLAST sequence bitscore bins (%)
>
> | Model\BitScore | [0,30] | (30,35] | (35,40] | (40,45] | (45,50] | (50,55] | (55,60] | (60,65] | (65,70] |
> |---|---|---|---|---|---|---|---|---|---|
> | Prot2Text | 6.58 | 6.41 | 7.51 | 8.63 | 12.58 | 16.03 | 20.76 | 21.68 | 27.04 |
> | ProtT3 | 6.19 | 5.26 | 7.12 | 9.76 | 13.10 | 10.87 | 18.97 | **30.95** | **40.82** |
> | Prot2Text-V2 | **11.89** | **15.21** | **14.78** | **14.00** | **16.05** | **22.20** | **28.65** | 29.26 | 31.08 |
>
> Table 3: BioBert F1 Score of ProtT3 across BLAST sequence bitscore bins (%)
>
> | Model\BitScore | [0,30] | (30,35] | (35,40] | (40,45] | (45,50] | (50,55] | (55,60] | (60,65] | (65,70] |
> |---|---|---|---|---|---|---|---|---|---|
> | Prot2Text | 73.38 | 73.92 | 74.30 | 74.11 | 75.04 | 76.64 | 78.24 | 78.31 | 78.98 |
> | ProtT3 | 67.57 | 66.44 | 69.19 | 74.37 | **79.73** | 78.11 | 76.44 | 81.06 | **83.99** |
> | Prot2Text-V2 | **76.74** | **77.23** | **77.52** | **77.68** | 78.99 | **80.61** | **81.07** | **81.95** | 83.49 |
>
> These two additional tables further prove that Prot2Text-V2 outperforms both Prot2Text and ProtT3 across nearly all low-homology bins, especially in the [0,40%] range in both BLEU-4 F1 score and BioBERT F1 score.
>
> However, we restate that the original train-test split of the ProtT3 paper is strongly imbalanced and lacks attention to the low-homology proteins. Although our test set contains 60% fewer proteins overall, it includes more than twice as many low-homology proteins compared to theirs.
>
> **[W3 on P2T-GPT]**
>
> We thank the reviewer for pointing out the relevant paper, P2T-GPT. While P2T-GPT trains its encoder and GPT decoder from scratch, our work explores the opposite direction, leveraging large pretrained models with novel contrastive learning scheme to maximize transfer learning and scalability.
>
> Table 4: Comparison between Prot2Text-V2 and P2T-GPT
> | model | Exct | B-2 | B-4 | R-1 | R-2 | R-L | RBT | BBT |
> |---|---|---|---|---|---|---|---|---|
> | P2T-GPT | 17.24 | 33.30 | 26.06 | 38.14 | 23.72 | 35.47 | 88.23 | 78.66 |
> | Prot2Text-V2 | **39.16** | **46.67** | **43.34** | **57.24** | **50.17** | **55.39** | **91.95** | **86.81** |
>
> P2T-GPT is a relatively lightweight model (~160M parameters) trained entirely from scratch without leveraging large-scale pretrained components or proper multimodal alignment. We conclude from these results that Prot2Text-V2 benefits from integrating powerful pretrained models (ESM and LLaMA) and an H-SCALE adapter, enabling it to capture richer biological and linguistic priors. This architectural difference is reflected in the substantial performance gains across all evaluation metrics, especially in semantic fidelity and functional relevance. We still find this paper of great value as it pioneered in addressing the problem in a novel approach at that time. We will make sure to add these comparison results and cite the paper in the camera-ready version.

---

> > ### Comment · Reviewer_4fdJ · 2025-08-07
> >
> > Thank you, authors, for providing a detailed rebuttal. The experimental results presented by the authors are convincing, for example, the tables in W1.3, W2, and W3. Regarding novelty, I acknowledge the authors' claim that their method differs from previous approaches. However, it seems more like a combination of several existing techniques. I'm not underestimating the contribution of this work, but I think the novelty is still relatively limited. Taking everything into account, since the authors have provided more experimental results, including comparisons to ProtT3 on low-homology and P2T-GPT, I would prefer to give a result that is somewhere between a weak reject and a weak accept, leaning towards acceptance.

---

> > > ### Author Response · Authors · 2025-08-07
> > > **Response to Reviewer 4fdJ**
> > >
> > > Dear reviewer:
> > >
> > > Thank you very much for your thoughtful and positive feedback. We really appreciate your acknowledgement of the strength of our experimental results. We are very glad that the clarifications about how our work differs from previous research helped address your concerns about novelty.
> > >
> > > It’s great to hear that our rebuttal improved your overall impression of the paper. Your feedback was incredibly helpful in making our work clearer and sharper. We sincerely thank you for your time and consideration.

---

### Official Review · Reviewer_6AtP · 2025-07-02

**Clarity:** 3
**Significance:** 3
**Originality:** 3
**Rating:** 4
**Confidence:** 4

**Summary:**

The paper introduces Prot2Text-V2, a multimodal framework for generating natural language descriptions of protein functions directly from amino acid sequences. Existing methods rely on structured ontologies or homology, limiting flexibility and generalization to novel proteins. Prot2Text-V2 addresses this by combining the ESM-3B protein language model encoder with an LLaMA-3.1-8B-Instruct decoder, connected via a nonlinear modality projector. A key innovation is Hybrid Sequence-level Contrastive Alignment Learning (H-SCALE), which aligns protein and text embeddings using mean and standard deviation pooling to enhance cross-modal consistency. After contrastive alignment, the model is fine-tuned with instruction-based LoRA to generate functional descriptions. Experiments on 250K SwissProt entries show Prot2Text-V2 outperforms baselines in low-homology scenarios, with improvements in lexical (e.g., BLEU-4: 43.34%) and semantic metrics (e.g., GPT score: 64.58).

**Questions:**

what kind of wet-lab validation experiments of selected proteins  do you want to do?

**Ethical Concerns:**

["NO or VERY MINOR ethics concerns only"]

**Final Justification:**

The authors give the extra results with various model sizes.

**Limitations:**

Yes. They have.

**Paper Formatting Concerns:**

No.

**Quality:**

3

**Strengths And Weaknesses:**

Strengths:

H-SCALE Contrastive Learning: Aligns protein and text embeddings using global and local features, improving semantic consistency.

Multimodal Architecture: Bridges protein sequences and natural language via ESM-3B and LLaMA-3.1, enabling free-form functional annotation.

Weakeness:

Training requires 8x A100 GPUs and 50 hours, restricting accessibility for small labs. LLM decoder (8B parameters) increases inference latency compared to lighter models.

---

> ### Author Rebuttal · Authors · 2025-07-31
>
> Dear reviewer:
>
> Thank you for your positive and encouraging review of our paper. We are delighted that you recognized the strengths of our work, particularly the Prot2Text-V2 multimodal architecture and the H-SCALE contrastive learning method.
> We address your questions regarding the computational complexity and the wet lab experiments below:
>
> **[Weakness on Computational Complexity]**
>
> We recognize that the training cost (8× A100 GPUs for ~50 hours) could be a hurdle for smaller labs, but it is still within a reasonable range compared to other standard LLMs that require thousands of GPU hours. While our current focus has been on demonstrating the feasibility and performance of protein captioning using large-scale models, model efficiency is an important future direction. To that end, based on your insightful comment, we trained smaller variations of our model and we aim to release the pretrained checkpoints to support reproducibility and allow researchers to build upon our work without retraining from scratch.
>
> We present the results of the smaller models in the table below. These models use a reduced decoder size and a lighter backbone encoder, making them more suitable for resource-constrained settings.
> Despite their smaller size, they achieve competitive performance and outperform the other baseline methods. Nonetheless, our largest model yields the best overall results.
>
> Table 1: Ablation results with various model sizes
>
> | Prot2Text-V2 ablations | Exct | B-2 | B-4 | R-1 | R-2 | R-L | RBT | BBT | latency |
> |---|---|---|---|---|---|---|---|---|---|
> | Prot2Text (898M) | 32.05 | 26.26 | 32.45 | 50.49 | 42.60 | 48.33 | 90.56 | 84.26 | - |
> | ESM-150M + LLaMA-3.2-1B | 33.25 | 40.61 | 37.26 | 54.69 | 47.12 | 52.93 | 90.99 | 85.68 | **0.73s** |
> | ESM-650M + LLaMA-3.2-3B | 35.37 | 43.54 | 40.11 | 56.52 | 49.38 | 54.80 | 91.30 | 86.17 | 1.32s |
> | ESM-3B + LLaMA-3.1-8B | **39.16** | **46.67** | **43.34** | **57.24** | **50.17** | **55.39** | **91.95** | **86.81** | 2.03s |
>
> Also, even our largest model with the 8B LLM decoder can perform inference in less than 2 seconds, which does not pose a significant hurdle.
>
> **[Question on Wet-lab Experiments]**
>
> Our goal is to validate whether the generated functional descriptions from Prot2Text-V2 are biologically meaningful and experimentally actionable. To that end, we envision the following wet-lab validation experiments:
>
> * In vitro expression and solubility screening:
> We will express selected proteins in heterologous systems (e.g., E. coli, HEK293) to assess expression yield and solubility, which are often indicative of proper folding and biochemical tractability.
>
> * Functional assays:
> Predicted enzymatic or molecular activities (e.g., ATPase, kinase, transport activity) will be tested using established in vitro biochemical assays to confirm functional relevance.
>
> * Subcellular localization:
> Proteins will be tagged (e.g., GFP fusion) and imaged using fluorescence microscopy to test localization predictions, such as mitochondrial or membrane targeting.
>
> * Binding and interaction studies:
> To evaluate predicted protein–protein or protein–ligand interactions, we will use affinity chromatography, pull-down assays, or surface plasmon resonance (SPR) to detect binding specificity and affinity.
>
> These steps are non-trivial and resource-intensive, but are part of future efforts to use protein captioning for guiding experimental protein function discovery.

---

> > ### Comment · Reviewer_6AtP · 2025-08-04
> >
> > Thanks for the rebuttal. Thanks for giving the extral  ablation results with various model sizes.

---

> > > ### Author Response · Authors · 2025-08-04
> > > **Response to Reviewer 6AtP**
> > >
> > > Dear reviewer:
> > >
> > > Thank you for your kind reply and for taking the time to review our additional results. We're glad the ablation studies were helpful, and we truly appreciate your constructive feedback throughout the process.

---

### Official Review · Reviewer_KRBB · 2025-07-23

**Clarity:** 3
**Significance:** 2
**Originality:** 2
**Rating:** 4
**Confidence:** 3

**Summary:**

The paper tackles the task of protein captioning - given an amino acid sequence, predict a textual description with the hypothesized protein function. The paper proposes a new model which leverages a protein encoder and an LLM decoder linked via a modality projector. The model is trained via a two step training process: a contrastive learning step aimed to align the latent representations of protein sequences and of textual descriptions (H-SCALE), and a fine-tuning step aimed to guide the LLM decoder to produce protein textual descriptions. The approach is evaluated via multiple evaluation metrics on held out protein subsets of various sequence similarity to the training set, and compared to baselines to show improved results across held out sets and metrics. The claimed contributions are 1) the overall Prot2Text-V2 model, 2) the H-SCALE contrastive alignment training procedure, 3) the approach’s strong performance compared to baselines in the low homology regime.

**Questions:**

H-SCALE
* From what I understand, the ProtT3 baseline the paper mentions is also using an ESM encoder with an LLM decoder, a two step training process with a contrastive learning step followed with a protein captioning training step. The major difference flagged in the paper is Q-Former vs. H-SCALE. Can the differences in performance reported in Table 2 be attributed primarily to Q-Former vs. H-SCALE or are there other differences between the two approaches as well?
* H-SCALE is posed as a new contrastive learning scheme that also leverages standard deviation and intermediate representations. Have you considered quantifying how much these choices impact alignment quality (in the style of Figure 4) or downstream performance when using sequence embeddings only?
* Figure 6 highlights how the model that was trained with H-SCALE converges faster (and to a higher final performance) as a function of the number of training steps performed. Does this account for the steps performed during contrastive learning?

Overall performance
* Can you please explain the difference between the BLEU-2 (B-2) score in Table 1 and the BLEU-2 F1 Score (%) in Figure 5? What made you choose different metrics in Figure 5? Also, what made you exclude the exact match performance in Figure 6?
* Are all the baselines included in the main text included in the human expert evaluation? I did not see BLAST and ProtT3. Did you find the human expert assessments to be correlated with the metrics used in the main text?

**Ethical Concerns:**

["NO or VERY MINOR ethics concerns only"]

**Limitations:**

Yes

**Paper Formatting Concerns:**

No concerns

**Quality:**

3

**Strengths And Weaknesses:**

Strengths
- The paper tackles the task of annotating protein sequences with textual descriptions which is an interesting problem in protein modeling. The proposed approach (pretrained protein encoder, pretrained llm decoder, modality projector, two-step training approach, H-SCALE) is clearly described and the authors add intuitions throughout the text for various architectural choices
- The overall approach shows improved performance across several evaluation settings including held out protein subsets of various sequence similarity to the training set and multiple evaluation metrics

Weaknesses:

Overall, the originality and significance seems modest and I recommend a borderline accept due to the improved results. In particular, the task of protein captioning and the train-test splits used are established in cited prior work (Prot2Text, ProtT3) and the paper proposes a new approach that improves performance across metrics:
- Approach: The paper notes the similarity to ProtT3 and highlights the H-SCALE contrastive learning approach as the notable difference. This seems like an incremental improvement to me, but I asked a few questions below that could help me understand the significance of this contribution more and I will reassess this after rebuttal
- Experimental setting: I think the paper's originality could be improved by describing whether the analysis/metrics are newly introduced or mimic Prot2Text.
- Results: I think the paper's significance could be improved by providing more insights into the results, e.g. discussing the discrepancy between metrics in Figure 5, whether the metrics highlighted in the main text correlate to the human expert evaluation described in the appendix, etc.

---

> ### Author Rebuttal · Authors · 2025-07-31
>
> Dear reviewer:
>
> Thank you for your thorough review and positive feedback on our work. We appreciate that you found our intuition well explained, implementation clearly described, the protein captioning task meaningful, and our results showing improved performance to be a key strength. We are grateful for your insightful questions, which greatly helped us identify areas to strengthen in our manuscript. We address your specific concerns below.
>
> **[Weakness-1 (W1) and Question-1 (Q1) on Novelty]**
>
> While we build upon established components like ESM and LLMs, we believe that our proposed model Prot2Text-V2 has significant and novel methodological improvements over existing multimodal LLM designs. While there are some similarities with ProtT3 architecture, our model departs in the following essential respects:
>
> * ProtT3 relies on a Q-Former, an additional attention module with learnable queries of a fixed number and a redundant text encoder, to align the protein and text modalities. In contrast, Prot2Text-V2 follows a decoder-only design with a lightweight projector and with a parameter-free alignment method.
>
> * Rather than learning alignment implicitly through a query-attention module, H-SCALE directly aligns the projected protein representations to intermediate text embeddings with a new contrastive learning objective. This eliminates the learned bottleneck, preserves the full expressivity of the pretrained protein encoder, and avoids added parameters, latency, and memory, and also allows us to use a more powerful LLM decoder.
>
> * We also want to emphasize that contrastive learning from intermediate LLM embeddings is not trivial, and it has not been done in any prior work, since all previous methods use an additional text encoder.
>
> **[W2 on Originality]**
>
> Thank you for raising this important point. While our core evaluation protocol is compatible with prior work (e.g., BLEU, ROUGE etc), we extend it with key additions that introduce more biologically meaningful semantic evaluation signals, going beyond what was used in Prot2Text or ProtT3.
>
> First, we applied additional evaluations and analyses:
>
> * LLM-as-a-Judge: Protein function descriptions can be complex and semantically rich as models can produce valid but lexically novel captions. By leveraging LLMs generalization and domain knowledge, we can evaluate semantic fidelity and biological coherence. We use GPT-4 as a judge, rating the fluency and the functional correctness of the predictions.
>
> * Human Expert Evaluation: We conducted a systematic human evaluation involving domain experts who assessed model outputs along similar dimensions (fluency, function, plausibility). The evaluation is conducted blindly across the multiple models.
>
> Second, we introduced a novel bitscore-based evaluation benchmark for low-homology proteins, which divides highly dissimilar proteins into finer-grained groups than previous approaches. Unlike identity scores, which only capture raw sequence similarity, bitscore reflects the statistical significance of alignments, incorporating both alignment quality and length. This makes it more sensitive and informative for detecting remote homologs, enabling more precise and meaningful evaluation. In contrast, identity scores often fail to distinguish between weak true homology and random noise, especially in low-similarity regions.
>
> Third, we further investigated the performance of the prediction model when the name and taxonomical information of the protein is provided. The analysis in Appendix F shows that including names boosts performance, probing the upper bound of the model’s capacity with additional information.
>
> **[W3 on Result Insights]**
>
> Even though there are small discrepancies in the metrics, we still observe a consistent trend, which shows that our approach outperforms the baselines on the low-homology settings across all the metrics.  Regarding the small discrepancies observed in Figure 5, it likely stems from the varying sensitivity of the metrics to different aspects of model performance (precision vs. coverage). The thresholds (where the retrieval-based BLAST predictor begins to outperform our model) on semantic metrics are generally higher than on lexical ones, as shown in Figure 5. This suggests that our LLM-based model is capable of generating accurate descriptions, albeit in a different style than the ground truth annotations, capturing the correct semantics while exhibiting variability in lexical choices. We observed a strong alignment between GPT-as-a-judge scores and human expert evaluations, suggesting that both metrics can better handle the diversity and paraphrasing, measuring the true performance of the model beyond surface-level similarity. We appreciate the suggestion and will incorporate this analysis into the final version, along with an expanded discussion of Figure 5 to improve clarity and aid interpretation.
>
> **[Q2 on Alignment Quality]**
>
> You raised an excellent point about comparing the impact of different alignment techniques on the model. Our motivation is that the average pooling captures the global semantic similarity, whilst the standard deviation pooling (a diagonal approximation to the covariance) captures the local variation and dispersion. Unfortunately, different alignment techniques are designed with different training losses, and a direct quantitative comparison is hardly feasible or meaningful.
>
> However, we can observe the impact of each design choice on our downstream protein function prediction task. Based on your insightful suggestion, we performed an ablation study comparing H-SCALE with various baselines. We present the results below:
>
> Table 1: Ablation study of Prot2Text-V2 with Different Contrastive Learning Schemes
>
> | **Ablations** | **Exct** | **B-2** | **B-4** | **R-1** | **R-2** | **R-L** | **RBT** | **BBT** |
> |---|---|---|---|---|---|---|---|---|
> | w/o alignment | 36.52 | 42.70 | 38.98 | 54.71 | 47.62 | 52.90 | 91.33 | 85.58 |
> | w/ mean-pooling alignment | 37.01 | 44.39 | 41.23 | 55.30 | 48.27 | 53.62 | 91.61 | 86.19 |
> | w/ std-pooling alignment | 33.42 | 37.28 | 33.78 | 50.49 | 43.22 | 48.75 | 90.93 | 84.70 |
> | w/ H-SCALE | 39.16 | 46.67 | 43.34 | 57.24 | 50.17 | 55.39 | 91.95 | 86.81 |
>
> **[Q3 on Convergence Speed]**
>
> Figure 6 does not include the preliminary alignment phase but only presents the results from supervised fine-tuning over approximately 48 hours. However, our H-SCALE contrastive learning converges in just 10 hours on the same computing platform. The accelerated convergence in the second training stage offsets the cost of the initial alignment phase. We will make sure to clarify this in the final version of the paper. That said, the primary advantage of H-SCALE lies not just in training efficiency, but in its improved final performance,
>
> **[Q4 on Evaluation Metrics]**
>
> > “the BLEU-2 (B-2) score in Table 1 and the BLEU-2 F1 Score (%) in Figure 5”
>
> Thank you for pointing it out. All previously reported BLEU-2 values should be interpreted as the BLEU-2 F1 score, and all the metrics are the same in Table 1 and Figure 5. We will correct it in the final version.
>
> Additionally, for Figure 6, although exact match ratio, rouge scores etc are valuable to properly evaluate the performance, we consider Bleu-4 F1 score as a direct, simple, and comprehensive way to trace the evolution of the training process. The exact match ratio can be too harsh on minor variations and is sensitive to punctuation or formatting, and thus cannot properly trace the evolution. However, for completeness, we will include all the metrics in the Appendix for the camera-ready version.
>
> **[Q5 on Human Evaluation]**
>
> > “baselines included in the main text included in the human expert evaluation”
>
> We are very grateful for your suggestion. We asked human expert evaluators to complete the evaluation matrix and we have the results from one of them until now. We expect to have the results from the second reviewer in the discussion phase. We present the results below, where Prot2Text-V2 (3.61) scored higher than ProtT3 (2.78) and BLAST (3.22) in expert evaluation. This reinforces that our model generates more semantically and biologically meaningful captions than both template-based and generative baselines.
>
> Table 2: Human expert evaluation on ProtT3 and BLAST retrieval-based predictor
>
> | Protein\Model | PTT3 | BLAST |
> |---|---|---|
> | Q9H2D6 | 1 | 4 |
> | B2RTY4 | 3 | 2 |
> | Q29537 | 4 | 5 |
> | Q32KY4 | 4 | 5 |
> | P00533 | 2 | 1 |
> | P04806 | 4 | 5 |
> | O14929 | 2 | 2 |
> | P00846 | 4 | 4 |
> | P06213 | 1 | 1 |
> | **Avg Score** | **2.78** | **3.22** |
>
> Also, based on your insightful suggestion, we computed the Pearson correlation of the metrics from Table 1 with the expert evaluations. The Pearson correlations with expert evaluations show that while BLEU and ROUGE have moderate alignment (~0.60), BioBERT (0.68) and GPT score (0.73) correlate more strongly, likely due to their deeper semantic understanding in the biological domain.
>
> Table 3: Pearson correlation coefficient among automatic and human evaluation metrics
>
> | Metric  | Bleu2 | Bleu4 | Rouge | Roberta | BioBert | GPT |
> |---|---|---|---|---|---|---|
> | **Human** | 0.63 | 0.61 | 0.59 | 0.55 | 0.68 | 0.73 |

---

> > ### Author Response · Authors · 2025-08-04
> > **Follow-up by Authors Regarding Human Evaluation Results**
> >
> > Dear reviewer:
> >
> > With input from a second human expert, we have now completed Table 2 for the human evaluation. The results  remain consistent, as Prot2Text-V2 (3.61) continues to outperform both ProtT3 (3.06) and BLAST (3.22) baselines.
> >
> > | Protein\Model | PTT3 | - |  | BLAST | - |
> > |---|---:|---|---|---:|---|
> > | Q9H2D6 | 1 | 1 |  | 4 | 5 |
> > | B2RTY4 | 3 | 3 |  | 2 | 2 |
> > | Q29537 | 4 | 3 |  | 5 | 3 |
> > | Q32KY4 | 4 | 5 |  | 5 | 4 |
> > | P00533 | 2 | 2 |  | 1 | 1 |
> > | P04806 | 4 | 5 |  | 5 | 5 |
> > | O14929 | 2 | 5 |  | 2 | 3 |
> > | P00846 | 4 | 5 |  | 4 | 5 |
> > | P06213 | 1 | 1 |  | 1 | 1 |
> > | **Avg Score** | **3.06** | - |  | **3.22** | - |
> >
> > We look forward to your response and sincerely thank you for your continued engagement.

---

### Note · Authors · 2025-08-12

Honorable Chairs and Reviewers,

We sincerely thank all reviewers for their positive feedback and valuable insights. We are pleased that all the reviewers liked our contribution and posed comments highlighting the strengths of our manuscript:

- Addressing the underexplored free-text protein function prediction task;
- Introducing the novel Prot2Text-V2 architecture and the H-SCALE algorithm, which greatly improve semantic alignment consistency;
- Providing a clear description of our idea and its underlying intuition;
- Achieving SOTA performance across multiple metrics in low-homology settings.

In the revised manuscript, building on the fruitful discussions with the reviewers, we will incorporate the additional studies and comparisons to further enhance clarity and emphasize the strengths of our approach, including:

- Additional experiments proving that we consistently outperform ProtT3 in low-homology domains, with especially strong gains in the most challenging low-similarity cases;
- Validation of the design choices through an extensive ablation study, revealing that aligning both first and second moments delivers the highest accuracy and robustness;
- Expanded theoretical motivation and discussion, offering deeper intuition and clarity into the approach’s design and strengths;
- A clearer articulation of our novel contributions, showing how our H-SCALE forms a new fine-tuning framework tailored to the unique challenges of multimodal protein function prediction.

Additionally, we respectfully emphasize that our study goes well beyond simply assembling existing tools. By integrating them into a novel framework with a new alignment algorithm, we tackle the challenging and underexplored problem of protein function prediction in low-homology settings. Through rigorous validation, our method consistently outperforms current SOTA approaches. It also generalizes well to unseen proteins, addressing a limitation that has slowed progress in this area for years.

Finally, we greatly appreciate the reviewers’ recognition of this work’s value. We believe that all questions and concerns raised by the reviewers are addressed with detailed, evidence-based responses, ensuring that each point was met with clarity and substantive reasoning. We are confident that our results represent a meaningful step forward, with clear potential to improve protein function prediction in realistic low-homology scenarios and to open new directions for future work.

---

### Decision · Program_Chairs · 2025-09-17

**Decision:**

Accept (poster)

**Comment:**

This paper introduces Prot2Text-V2, a multimodal framework for generating natural language descriptions of protein functions directly from amino acid sequences. A key innovation is the hybrid sequence-level contrastive alignment learning. The model uses a novel contrastive method by combining ESM and LLAMA. While the originality is modest as it primarily combines existing components, this method shows consistent improvements. All reviewers vote for acceptance, recognizing the practical significance, despite incremental novelty concerns.